# Sequences of Logits Reveal the Low Rank Structure of Language Models

**Noah Golowich**[*]
Microsoft Research
nzg@cs.utexas.edu

**Allen Liu**[*]
UC Berkeley
aliu42@berkeley.edu

**Abhishek Shetty**[*]
MIT
shetty@mit.edu

## Abstract

A major problem in the study of large language models is to understand their inherent low-dimensional structure. We introduce an approach to study the low-dimensional structure of language models at a model-agnostic level: as sequential probabilistic models. We first empirically demonstrate that a wide range of modern language models exhibit *low-rank* structure: in particular, matrices built from the model's logits for varying sets of prompts and responses have low approximate rank. We then show that this low-rank structure can be leveraged for generation — in particular, we can generate a response to a target prompt using a linear combination of the model's outputs on *unrelated, or even nonsensical* prompts.

On the theoretical front, we observe that studying the approximate rank of language models in the sense discussed above yields a simple *universal abstraction* whose theoretical predictions parallel our experiments. We then analyze the representation power of the abstraction and give provable learning guarantees.

## 1 Introduction

Understanding the structure of language has been a long-standing goal in computer science, motivating various fundamental models (Shannon, 1951; Chomsky, 2002) such as Hidden Markov Models (HMMs), finite state automata, and formal grammars (see (Jurafsky & Martin, 2025)).

Such questions have taken on a new light in recent years with the advent of Large Language Models (LLMs); despite extensive study, though, their success at modeling language has largely eluded a mathematically rigorous understanding. Such an understanding, while of interest in its own right, is also crucial when it comes to evaluating the possible risks associated with LLMs. Indeed, a sizeable literature has uncovered surprising ways to bypass the safety mechanisms of LLMs (e.g., (Wei et al., 2023; Zou et al., 2023)) as well as ways they can become misaligned during training (Razin et al., 2024; Betley et al., 2025). Understanding and defending these vulnerabilities often requires mathematical insight into the model's workings (Elhage et al., 2021; Olsson et al., 2022; Turner et al., 2023; Arditi et al., 2024).

One of the principal ingredients we still lack when it comes to obtaining a more solid theoretical foundation for studying LLMs are *simple universal abstractions* that are realistic enough to make testable predictions about deployed LMs and are mathematically tractable enough to support provable guarantees. Many such models have been proposed, ranging from simple types of transformers (Sanford et al., 2024), to low-depth circuits (Merrill & Sabharwal, 2023). While such frameworks have led to important insights, e.g., regarding expressivity, most of them are limited in the types of models they can represent (e.g., transformers of a certain depth (Merrill & Sabharwal, 2025)) or to certain types of tasks (e.g., RL fine-tuning (Foster et al., 2025)). Further, they are often not very capable of making precise predictions about the structure and behavior of modern LLMs.

In this paper, inspired by the folklore belief that language possesses intrinsic low-dimensional structure, we propose an *architecture-agnostic framework* which allows us to study how information in language is represented in low-dimensional subspaces. In particular, we study *logit matrices* associated to any language model (Section 1.1). This approach treats the language model as a sequential

---

[*]Equal contribution.

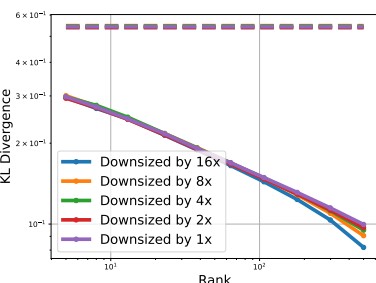

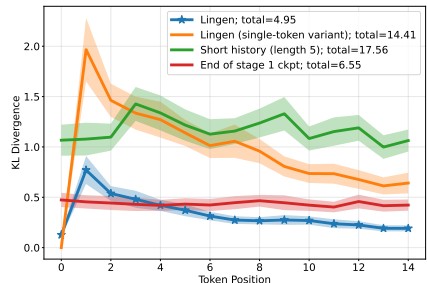

(a) OLMo-7b low-rank approximation        (b) Performance of LINGEN with OLMo-1b

Figure 1: **(a)**: Low-rank approximation error (measured by *average KL divergence*; see Theorem 3.1) of the extended logit matrix for OLMo-7b, and ranks 5-500. For fixed sets $\mathcal{H}, \mathcal{F}$, the approximation errors for the logit matrix $\mathcal{L}_M(\mathcal{H}, \mathcal{F})$ behave according to a similar power law as to those of various sub-matrices with $\{2, 4, 8, 16\}$-times fewer entries. Dashed line at top shows performance of a (suboptimal) rank-1 baseline. **(b)**: Performance of our generation procedure (LINGEN; star markers) which exploits the low-rank structure of the extended logit matrix to generate from a given "target" prompt by only querying the language model on nonsensical prompts unrelated to the target. We plot the KL divergence between LINGEN and the true language model (OLMo-1b) at each token position, as well as various baselines (solid lines; see Section 3.3).

probabilistic map from sequences to sequences, sidestepping architecture-specific details. A starting point for our work is the observation that the logit matrices associated to modern autoregressive LLMs are well-approximated by low-rank matrices (Section 3.1). Additionally, this low-rank structure can be exploited to exhibit several surprising phenomena, such as the ability to generate from a given model by only querying it on unrelated or even nonsensical prompts (Sections 3.2 and 3.3). Moreover, such a low-rank assumption yields a rich theoretical landscape, from both a learning-theoretic and generative modeling standpoint (Section 4).

## 1.1 STRUCTURE ARISES FROM THE EXTENDED LOGIT MATRIX

Our goal is to study low-dimensional representations of language models: *what aspects of the model should such a representation capture?* One natural requirement is that it allows us to compress the past in order to generate the future. Perhaps the most fundamental and extensively-studied type of such compressions are those which are *linear*, in the following sense. For a sequence of tokens $h$ (e.g., a prompt), the compression consists of a vector $\phi(h) \in \mathbb{R}^d$ so that the probability of observing any possible completion $f$ (e.g., a response) is proportional to $\exp(\langle\phi(h), \psi(f)\rangle)$, for some embedding vector $\psi(f) \in \mathbb{R}^d$ depending only on $f$.

We remark that a very similar setup has been considered in the context of LLMs, *when the sequence $f$ consists of a single token*: for essentially all modern LLMs, due to the structure of the unembedding matrix, the probability of observing a *single token* $y$ following a sequence $h$ is proportional to $\exp(\langle\phi(h), \psi(y)\rangle)$, for some embedding maps $\phi, \psi$. This property has been termed the *softmax bottleneck* (Yang et al., 2017), and has had a broad array of implications, ranging from expressivity and architecture (Press & Wolf, 2016; Kanai et al., 2018; Ganea et al., 2019; Chang & McCallum, 2022; Godey et al., 2024) to security (Carlini et al., 2024; Finlayson et al., 2024). However, this observation only allows us to reason about one token in the future. The key insight of our work is that this low-dimensionality persists, even when we consider logits over longer sequences of tokens.

To study this low-dimensionality over longer sequences, we introduce the *extended logit matrix* as our main object of study. Its rows are indexed by all possible sequences of tokens, which we refer to as *histories*. Roughly speaking, its columns are also indexed by all possible sequences, to be interpreted as possible *completions* of a history; we refer to them as *futures*. In essence, the entry of the extended logit matrix corresponding to a given $(h, f)$ pair and a language model $M$ is given by $\log \Pr_M[f \mid h]$, i.e., the logarithm of the probability that $M$ assigns to $f$ given context $h$. This

choice is natural given the above discussion— indeed, the fact that $\log \Pr_M[f \mid h] \approx \langle \phi(h), \psi(f) \rangle$ for some feature mappings $\phi, \psi$ (as discussed above), is equivalent to the following statement:

> The (extended) logit matrix is approximately low-rank for modern language models.

The main premise of this work is that the above statement is true for modern autoregressive LLMs. While we cannot construct the entire extended logit matrix (which is exponentially large), we can construct submatrices indexed by sets of histories $\mathcal{H}$ and futures $\mathcal{F}$, and measure how the low-rank structure of these submatrices (denoted $\mathcal{L}_M(\mathcal{H}, \mathcal{F})$) evolves with their size (Figure 1). As is shown, this approximate low-rank structure is remarkably consistent as we scale up the number of histories and futures. We emphasize both that this low-rank structure is atypical for a (random) matrix of the same dimensions and that it is *not* a consequence of the low-rank structure of the *single-token* logit matrix (see Section C). For the formal definition of the extended logit matrix, which differs slightly from the simplified description here, we refer the reader to Section 2.

### 1.2 OUR CONTRIBUTIONS

In this paper, we conduct a thorough empirical study of the extended logit matrix across both a wide range of LLMs and choices for the histories and futures, and complement our empirical findings with theoretical results on expressivity and learnability. Our framework suggests a number of intriguing open questions relating to interpretability, safety, and the theoretical underpinnings of LLMs.

**Empirical Findings: low-rank structure.** In Section 3.1, we study the extended logit matrices defined by a wide range of language models, and sources of histories and futures. We observe that the extended logit matrices are all well-approximated by low-rank matrices, and that the quality of the best low-rank approximation follows a power law which is strikingly consistent as the matrix is scaled up. We also observe that the low-rank structure is *not* present at the beginning of training, but rather emerges in the early stages of pre-training and further evolves throughout training.

**Empirical Findings: consequences of low-rank structure.** Given that the extended logit matrix is typically well-approximated by a low-rank matrix, a natural follow-up question is to understand the implied linear dependencies between the rows (i.e., histories) of the matrix. Such linear dependences may be seen as generalizations of classical examples such as boy − girl ≈ king − queen (Mikolov et al., 2013). In Section 3.2, we verify that these linear dependencies are consistent across the extended logit matrices of different LLMs. More surprisingly, they remain preserved even if we significantly change the futures (i.e., columns) of the matrix, *replacing the original futures with nonsensical futures consisting of random sequences of tokens*.

The fact that an LLM's low-rank structure is preserved even when prompted with nonsencial (random) sequences inspires us to investigate (in Section 3.3) the following setup for language generation. Given any *target prompt (i.e., history)*, using an appropriate extended logit matrix we can approximately write it as a linear combination of unrelated, even nonsensical, histories. We can then use this linear combination to generate a continuation to the target, by only querying the model on the unrelated histories. We find that this procedure generates coherent continuations, and its KL-divergence to the true model beats strong baselines, e.g., intermediate training checkpoints (Figure 1b). Such a procedure has the potential to circumvent defenses, such as prompt filters (Dong et al., 2024), established to ensure safety; we discuss further implications for AI safety in Section 3.3.

**Theoretical Framework.** In addition, a key aspect of our framework is that it supports a simple theoretical generative model that captures the low-rank structure that we find. Our empirical observations inspire us to revisit the Input Switched Affine Network (ISAN) of Foerster et al. (2017). ISANs were initially proposed as a simple, interpretable recurrent architecture, where a low dimension hidden state is updated through a linear transformation that is allowed to depend on the current sampled token, and were shown to achieve reasonable performance on language modeling tasks. We show that ISANs capture precisely those distributions that have (exact) low logit rank (Theorem 4.3).

To support the tractability and usefulness of ISAN as a theoretical model, we first study its representation power, showing that it can express state space layers, the key component of a variety of practically successful architectures broadly termed state space models (SSMs (Gu et al., 2021a; Gu & Dao, 2023; Gu, 2025)). We also show that it can express algorithmic behaviors like copying and noisy parity — the latter having implications for learnability. Since noisy parity is hard to learn

(Blum et al., 2003), efficiently learning an ISAN from samples is impossible in the worst case. In light of this, we give a provably efficient learning algorithm with *logit queries*, a setting that closely resembles practical model stealing setups for common APIs (Carlini et al., 2024).

**Related work.** We give a more detailed discussion of related work in Section A.

## 2 SETUP AND NOTATION

Our main object of study is autoregressive language models. We denote such models with the letter $M$ and the associated set of tokens with $\Sigma$. For a token $z \in \Sigma$ and a context sequence $y_{1:t} = (y_1, \ldots, y_t) \in \Sigma^t$, we will denote the corresponding probability distribution over the next token by $\Pr_M[\cdot \mid y_{1:t}]$, i.e., $\Pr_M[z \mid y_{1:t}]$ is the probability that the next token is $z \in \Sigma$. We let $\Sigma^t$ and $\Sigma^{\leq t}$ represent sequences of length $t$ and at most $t$, respectively, while $\Sigma^\star$ denotes the set of all sequences of tokens. Note that these include the empty string, which we denote by Null. We use $\circ$ to denote the concatenation of two sequences, i.e., $y \circ y'$ denotes the concatenation of $y$ and $y'$.

To formally introduce the key object that we study, the extended logit matrix, we first define the *mean-centered* logits; while empirically mean-centering doesn't make a large difference, it will be important for the theoretical generative model that we introduce later on (see Remark 1).

**Definition 2.1** (Mean-Centered Logits)**.** *Given a sequence* $y_{1:t} \in \Sigma^t$, *we define the* mean-centered *logits as* $L_M[z|y_{1:t}] = \log \Pr_M[z|y_{1:t}] - \frac{1}{|\Sigma|} \sum_{z' \in \Sigma} \log \Pr_M[z'|y_{1:t}]$ *for* $z \in \Sigma$.

Given a model $M$ and sets $\mathcal{H}, \mathcal{F} \subset \Sigma^\star$ consisting of sequences of tokens, the *extended logit matrix* $\mathcal{L}_M(\mathcal{H}, \mathcal{F})$ has rows indexed by $\mathcal{H}$ and columns indexed by $\mathcal{F} \times \Sigma$; accordingly, we will write $\mathcal{L}_M(\mathcal{H}, \mathcal{F}) \in \mathbb{R}^{\mathcal{H} \times (\mathcal{F} \times \Sigma)}$. We refer to elements of $\mathcal{H}$ as *histories* and of $\mathcal{F}$ as *futures*. Formally:

**Definition 2.2** (Extended Logit Matrix)**.** *Fix a model* $M$. *Given subsets of histories* $\mathcal{H} \subset \Sigma^*$ *and futures* $\mathcal{F} \subset \Sigma^*$, *the associated* extended logit matrix $\mathcal{L}_M(\mathcal{H}, \mathcal{F}) \in \mathbb{R}^{\mathcal{H} \times (\mathcal{F} \times \Sigma)}$ *is defined for* $h \in \mathcal{H}$ *and* $(f, z) \in \mathcal{F} \times \Sigma$ *by*

$$\mathcal{L}_M(\mathcal{H}, \mathcal{F})_{(h,(f,z))} := L_M[z \mid h \circ f].$$

Schematically, for histories $\mathcal{H} = \{h_1, \ldots, h_H\}$ and futures $\mathcal{F} = \{f_1, \ldots, f_F\}$, the matrix $\mathcal{L}_M(\mathcal{H}, \mathcal{F})$ has one block of length $|\Sigma|$ for each history-future pair $(h_i, f_j)$ and that block contains the (mean-centered) next-token logits for $h_i \circ f_j$:

$$\mathcal{L}_M(\mathcal{H}, \mathcal{F}) = \begin{bmatrix} \{L_M[z \mid h_1 \circ f_1]\}_{z \in \Sigma} & \cdots & \{L_M[z \mid h_1 \circ f_F]\}_{z \in \Sigma} \\ \vdots & \ddots & \vdots \\ \{L_M[z \mid h_H \circ f_1]\}_{z \in \Sigma} & \cdots & \{L_M[z \mid h_H \circ f_F]\}_{z \in \Sigma} \end{bmatrix}.$$

In the remainder of this paper, we will refer to the extended logit matrix as simply the logit matrix.

**Relation to Simplified Definition in Section 1.1.** The definition above, while slightly modified from the definition introduced in Section 1.1, serves as a linear representation in almost the same sense. Observe that if we set $\mathcal{F} = \Sigma^{\leq T}$, then a row $\mathcal{L}_M(\{h\}, \mathcal{F})$ contains all of the information necessary to sample a continuation of $h$ of up to length $T$. This is because, from the information in this row, we can compute $\log \Pr[f|h]$ for any $f \in \Sigma^{\leq T}$ by breaking up $f = z_{1:t}$ into tokens and summing up the token-by-token log conditional probabilities

$$\log \Pr[f|h] = \log \Pr[z_t|h \circ z_{1:t-1}] + \cdots + \log \Pr[z_1|h].$$

If we ignore the normalizing constant and replace the mean-centered logits with normalized logits, then the above is actually a linear function of the entries in the row $\mathcal{L}_M(\{h\}, \mathcal{F})$. The main purpose of writing the extended logit matrix in this new form in Theorem 2.2 is that the logits can be more directly extracted from the model.

## 3 EXPERIMENTS

### 3.1 LOW RANK STRUCTURE

In this section, we evaluate the extent to which the logit matrices associated to modern LLMs (Theorem 2.2) are actually low-rank. To do so, we computed logit matrices $\mathcal{L}_M(\mathcal{H}, \mathcal{F})$ corresponding

to various models $M$ and sets $\mathcal{H}, \mathcal{F}$ of sequences. The sets $\mathcal{H}, \mathcal{F}$ were generated as follows starting from a dataset $D$, according to some parameter $n \in \mathbb{N}$. We sampled $2n$ sequences from $D$ and for each of the $2n$ sequences included a random subsequence of it in either $\mathcal{F}$ or $\mathcal{H}$ (so that each of $\mathcal{F}, \mathcal{H}$ ends up with $n$ subsequences). We studied a broad range of datasets $D$ and open-source models $M$; precise details regarding the choices of $D, M$ as well as $\mathcal{H}, \mathcal{F}$ may be found in Section B.1.1. For the figures shown in the main body of the paper, the dataset $D$ was always taken to be the `wiki` split of `olmo-mix-1124` (OLMo et al., 2024).

We considered various values of $n$, ranging up to $10^4$. Note that the full logits matrix $\mathcal{L}_M(\mathcal{H}, \mathcal{F})$ has $n^2 \cdot |\Sigma|$ entries, which is not feasible to store given that for many models (e.g., OLMo) $|\Sigma| \approx 10^5$. Thus, in our experiments we used instead a sub-matrix of $\mathcal{L}_M(\mathcal{H}, \mathcal{F})$, which we denote by $\mathcal{L}_{M,k}(\mathcal{H}, \mathcal{F})$, for some parameter $k$. The submatrix $\mathcal{L}_{M,k}(\mathcal{H}, \mathcal{F})$ is obtained by selecting a subset of the columns as follows: for each future $f \in \mathcal{F}$, we select the columns indexed by $(f, z)$ where $z$ is one of the $k$ most-likely tokens following $f$, i.e., for which $\Pr_M[z \mid f]$ is largest. In our experiments we took $k = 50$; in Section B.1.5 we also repeated some experiments for (a) $k = 200$ and (b) where a *random* subset of $k = 50$ tokens was selected for each $f$, and observed similar results. At a high level, $\mathcal{L}_{M,k}(\mathcal{H}, \mathcal{F})$ can be interpreted as containing information about the model's predictions for the most-likely next tokens for each possible context $h \circ f$ (with $h \in \mathcal{H}, f \in \mathcal{F}$). We measured the degree to which the resulting logit matrix $\mathcal{L}_{M,k}(\mathcal{H}, \mathcal{F})$ is well-approximated by a low-rank matrix in two ways, discussed below. For ease of notation, we omit the parameter $k$ (and ignore the distinction between $\mathcal{L}_M(\mathcal{H}, \mathcal{F})$ and $\mathcal{L}_{M,k}(\mathcal{H}, \mathcal{F})$) in our description of the experiments throughout this section.

**Method 1: measuring the singular value decay.** In Figure 2 (and Figure 7, which shows additional LLMs), we display the singular values of the logits matrix $\mathcal{L}_M(\mathcal{H}, \mathcal{F})$ for various choices of models $M$ and various sizes $n$ of the sets $\mathcal{H}, \mathcal{F}$, when the dataset $D$ was `wiki`. The singular values decay approximately according to a power law, in that the $i$th singular value $\sigma_i$ of the matrix $\mathcal{L}_M(\mathcal{H}, \mathcal{F})$ is approximately $C \cdot i^{-\alpha}$ for some $C, \alpha > 0$ (depending on $M$).

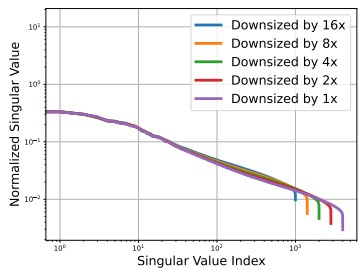

Figure 2: OLMo-7b singular values; Power law exponent $\alpha \approx 0.536$

As shown in Figures 2 and 7, the exponent $\alpha$ for most logit matrices is slightly greater than $1/2$; this holds as well for additional choices of the dataset $D$ (see Figure 9 in the appendix). Intriguingly, $\alpha = 1/2$ represents a *phase transition* for low-rank approximation, in the following sense (see Fact B.1 for a formal statement): if $\alpha > 1/2$, then for any constant $\varepsilon$, there is a *constant* $r_\varepsilon$ depending only on $\varepsilon$ so that a rank-$r_\varepsilon$ matrix $\varepsilon$-approximates the logit matrix. On the other hand, if $\alpha < 1/2$, then for sufficiently small constants $\varepsilon$, a rank *linear* in the dimension is needed.

When it comes to approximating the logit matrices of language models, we are most interested in the setting where $\mathcal{H}, \mathcal{F}$ contain *all* possible histories and futures (or at least those that are plausible in natural language), as then the logits matrix $\mathcal{L}_M(\mathcal{H}, \mathcal{F})$ contains *all the next-token prediction information* in the language model. In this setting, the logit matrix's dimensions are *exponential* in the length of the sequences, meaning that the case of power law decay with $\alpha < 1/2$ requires that the rank be exponentially large to achieve approximation error less than some absolute constant. This is in contrast to *constant* rank for the case $\alpha > 1/2$ which we observe in Figures 2, 7 and 9.

While it is infeasible to compute the logits matrix when $\mathcal{H}, \mathcal{F}$ are taken to be the sets of *all* histories and futures, it is apparent in the figures that the exponent of the power law remains essentially unchanged when we consider sub-matrices of $\mathcal{L}_M(\mathcal{H}, \mathcal{F})$ (modulo some degeneration of the power law at ranks that approach the size of $\mathcal{H}, \mathcal{F}$, which is expected). Thus, as $\mathcal{H}, \mathcal{F}$ **are scaled up, we expect that the same power law holds.**

**Method 2: measuring the approximation via KL divergence.** While measuring the decay of singular values of a matrix is mathematically convenient (in the sense of, e.g., Fact B.1), it does not directly yield a *probabilistic* interpretation of the fact that the logit matrix $\mathcal{L}_M(\mathcal{H}, \mathcal{F})$ is close to another (low-rank) matrix. To bridge this gap, we observe that any matrix $L \in \mathbb{R}^{\mathcal{H} \times (\mathcal{F} \times \Sigma)}$ (e.g., a logit matrix $\mathcal{L}_M(\mathcal{H}, \mathcal{F})$) induces a collection of $|\mathcal{H}||\mathcal{F}|$ "next token distributions" on $\Sigma$ by taking the softmax of the vector $L_{h,f} := (L_{h,(f,z)})_{z \in \Sigma}$, for each $h, f$. (Indeed, recall from Theorem 2.2

that when $L = \mathcal{L}_M(\mathcal{H}, \mathcal{F})$, we have $L_{h,f} = L_M[\cdot \mid h \circ f]$.) In place of Frobenius norm, we propose to measure the average KL divergence between these distributions:

**Definition 3.1** (Average KL divergence). *Fix sets $\mathcal{H}, \mathcal{F} \subset \Sigma^\star$, and consider matrices $L, A \in \mathbb{R}^{\mathcal{H} \times (\mathcal{F} \times \Sigma)}$. We define $D_{\mathsf{KL}}^{\mathsf{avg}}(L, A)$ to be the average KL divergence between the next-token distributions induced by the entries of $L_{h,f}$ and $A_{h,f}$ for each $(h, f)$ pair:*

$$D_{\mathsf{KL}}^{\mathsf{avg}}(L, A) = \frac{1}{|\mathcal{H}||\mathcal{F}|} \sum_{h \in \mathcal{H}, f \in \mathcal{F}} D_{\mathsf{KL}}(\mathrm{softmax}(L_{h,f}) \| \mathrm{softmax}(A_{h,f})).$$

Thus, for a low-rank matrix $A$, the quantity $D_{\mathsf{KL}}^{\mathsf{avg}}(\mathcal{L}_M(\mathcal{H}, \mathcal{F}), A)$ may be interpreted as the ability of the low-rank matrix $A$ to approximate the model $M$ in KL divergence, when restricted to contexts $h \circ f$ for $h \in \mathcal{H}, f \in \mathcal{F}$. It is straightforward to show (see Fact B.2) that we may bound

$D_{\mathsf{KL}}^{\mathsf{avg}}(\mathcal{L}_M(\mathcal{H}, \mathcal{F}), A) \le \frac{1}{|\mathcal{H}| \cdot |\mathcal{F}|} \cdot \|\mathcal{L}_M(\mathcal{H}, \mathcal{F}) - A\|_F^2$. Combining this fact, Fact B.1, and the observed decay in singular values in Figure 2, we should expect the average KL divergence between $\mathcal{L}_M(\mathcal{H}, \mathcal{F})$ and a rank-$r$ approximation to decay at least as fast as a power law (in $r$). This prediction is confirmed in Figure 1a (see also Figures 8 and 10 in the appendix), where we also observe consistency in the power law as the matrix size is increased, suggesting that such a **low-rank approximation holds when $\mathcal{H}, \mathcal{F}$ are even exponentially large.**

**Evolution of approximation during the course of training.** One intriguing exception to the findings discussed above is that for a "baseline" model $M$ corresponding to the "Step 0" checkpoint of OLMo-1b (i.e., before any training steps), the logit matrix does *not* appear to exhibit low-rank structure: see Figure 7d in the appendix, where we have $\alpha \approx 0.374$. Further, while the KL divergence to a low-rank approximation does decrease with the rank (Figure 8d), this decrease does not seem to follow a consistent power law as the matrix's size is increased, suggesting that for larger sets $\mathcal{H}, \mathcal{F}$ such a decrease may be diminished. These observations lead us to ask: *At what point in training does the logit matrix become (approximately) low rank?*

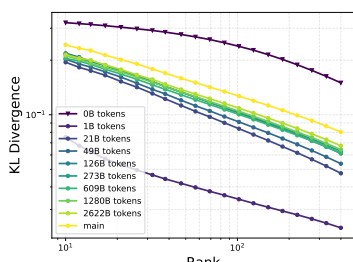

Figure 3: Low-rank approximations (wrt. avg. KL divergence) over Stage-1 pretraining of OLMo-1b.

To answer this question, we show, in Figure 3, the average KL divergence to a low-rank approximation for the logit matrices at various pre-training checkpoints of OLMo-1b (given a fixed set of $\mathcal{H}, \mathcal{F}$ each of size approximately 4000). Early in training, the KL divergence to a low-rank approximation drops significantly, then rises slowly towards its final value. This observation suggests many fascinating questions for follow-up work, foremost amongst them: **How and why does low-rank structure in the logit matrix emerge early in pre-training?**

## 3.2 SHARED STRUCTURE ACROSS MODELS & FUTURES

Next, we proceed to discuss some consequences of the (approximate) low-rank structure of the logit matrix $\mathcal{L}_M(\mathcal{H}, \mathcal{F})$. One of the most basic facts about low-rank matrices is that they have nontrivial (row) kernels, i.e., if a matrix $A \in \mathbb{R}^{\mathcal{H} \times (\mathcal{F} \times \Sigma)}$ (say, an approximation of $\mathcal{L}_M(\mathcal{H}, \mathcal{F})$) has rank much less than $|\mathcal{H}|$, then there is a large space of nonzero vectors $v \in \mathbb{R}^{\mathcal{H}}$ for which $v^\top \cdot A = 0$.

*What does the existence of such vectors mean?* Roughly speaking, such vectors $v$ represent "linear relationships" between different histories. Studying such linear relationships has been a mainstay of NLP over the last decade, with a simple example being relationships between *word embeddings* such as: boy $-$ girl $\approx$ king $-$ queen (Mikolov et al., 2013). *When it comes to logit matrices, do such linear relationships manifest in ways that correspond to semantic relations between histories?* Towards answering this question, we first describe the following "sanity check": for the sets of histories $\mathcal{H}$ and futures $\mathcal{F}$ considered in Section 3.1, for each $h \in \mathcal{H}$ we computed the $h' \ne h$ in $\mathcal{H}$ minimizing the norm of the difference between the corresponding rows of the logit matrix, namely $\mathcal{L}_M(\{h\}, \mathcal{F}) - \mathcal{L}_M(\{h'\}, \mathcal{F})$. Some samples of resulting pairs are shown in Table 1, where it can be seen that most of the pairs of histories share semantic similarities.

**Systematic evaluation: description.** We aim to more systematically evaluate whether vectors $v \in \mathbb{R}^{\mathcal{H}}$ describing linear relationships between rows (i.e., histories) of $A \approx \mathcal{L}_M(\mathcal{H}, \mathcal{F})$ are inherent to the histories themselves, as opposed to being spurious artifacts of either the model architecture or the set of futures $\mathcal{F}$. If indeed such vectors $v$ were *not* spurious, then we would expect that: *the induced linear relationship given by such vectors transfers (a) across the choice of futures $\mathcal{F}$ and (b) across models $M$.* In other words (regarding (a)): given distinct sets of futures $\mathcal{F}, \mathcal{F}'$ (drawn from different distributions), after choosing low-rank matrices $A, A'$ satisfying $A \approx \mathcal{L}_M(\mathcal{H}, \mathcal{F})$ and

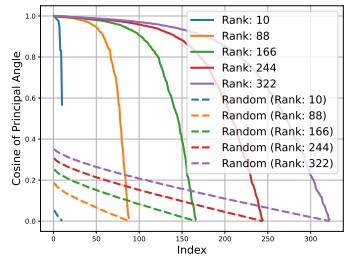

Figure 4: Cos of principal angles between column spaces of low-rank apx of $\mathcal{L}_M(\mathcal{H}, \mathcal{F})$ & $\mathcal{L}_M(\mathcal{H}, \mathcal{F}^{\mathrm{nonsense}})$.

$A' \approx \mathcal{L}_M(\mathcal{H}, \mathcal{F}')$, the row kernels of $A, A'$ should be roughly equal. This would mean that approximate linear relationships between the rows of $\mathcal{L}_M(\mathcal{H}, \mathcal{F})$ are also approximate linear relationships between the rows of $\mathcal{L}_M(\mathcal{H}, \mathcal{F}')$, and thus these *relations are between the histories themselves, independent of the choice of futures.*

Equivalently, since the column span of a matrix is the orthogonal complement of the matrix's row kernel, we expect that the column spans of $A, A'$ should be roughly equal. Moreover (regarding (b)), a similar conclusion should hold if $A, A'$ were low-rank approximations of $\mathcal{L}_M(\mathcal{H}, \mathcal{F})$ and $\mathcal{L}_{M'}(\mathcal{H}, \mathcal{F})$ for distinct *models* $M, M'$.

**Invariance to the choice of $\mathcal{F}$: "nonsense" futures.** To evaluate the above hypothesis regarding differing sets $\mathcal{F}, \mathcal{F}'$ of futures: for a fixed model $M$ (taken to be OLMo-1b), and sets $\mathcal{H}, \mathcal{F}$ as constructed in Section 3.1, we create a "nonsense" version $\mathcal{F}^{\mathrm{nonsense}}$ of $\mathcal{F}$ by randomly permuting all tokens amongst futures of $\mathcal{F}$. Given a parameter $r \in \mathbb{N}$, we let $A \in \mathbb{R}^{\mathcal{H} \times (\mathcal{F} \times \Sigma)}$ (resp., $A^{\mathrm{nonsense}}$) denote the best rank-$r$ approximation to $\mathcal{L}_M(\mathcal{H}, \mathcal{F})$ (resp., $\mathcal{L}_M(\mathcal{H}, \mathcal{F}^{\mathrm{nonsense}})$) in Frobenius norm. *Are the column spaces of $A, A^{\mathrm{nonsense}}$ approximately equal?* In Figure 4, we answer this by plotting, for various values of $r$, the (cosines of the) *principal angles* between the column spans of $A, A^{\mathrm{nonsense}}$.[1] A large fraction of the principal angles are close to $0$, i.e., have cosines close to $1$ (and this fraction is much larger than for a pair of independent uniformly random $r$-dimensional subspaces, which is shown in the dashed lines). This means that the column spaces of $A, A^{\mathrm{nonsense}}$ are mostly overlapping, and thus **linear relationships between histories transfer to significantly different sets of futures**.

The above experiment demonstrates that the model's logits $L_M[\cdot \mid h \circ f^{\mathrm{nonsense}}]$, for a history $h \in \mathcal{H}$ and a "nonsense" future $f^{\mathrm{nonsense}} \in \mathcal{F}^{\mathrm{nonsense}}$, still yield significant information about $h \circ f$, for real futures $f \in \mathcal{F}$. In particular, given a vector $v$ representing some linear relationship between histories in the sense that $v^{\top} \cdot \mathcal{L}_M(\mathcal{H}, \mathcal{F}^{\mathrm{nonsense}}) \approx 0$, we typically have also $v^{\top} \cdot \mathcal{L}_M(\mathcal{H}, \mathcal{F}) \approx 0$.

**Invariance to the choice of $M$.** In Figure 12 in the apppendix, we test the additional hypothesis of whether the linear relationships between histories are shared across different models. We observe a similar overlap between the column spaces of low-rank approximations to $\mathcal{L}_M(\mathcal{H}, \mathcal{F})$ for various pairs $(M, M')$ of distinct models. This means that, at least to some extent, *these linear relationships are inherent to the sentences themselves, independent of the choice of model.*

### 3.3 EXPLOITING LOW RANK FOR GENERATION

Finally, we investigate one consequence of the fact that there are "linear relationships" between histories as captured by vectors $v \in \mathbb{R}^{\mathcal{H}}$ satisfying $v^{\top} \cdot \mathcal{L}_M(\mathcal{H}, \mathcal{F}) \approx 0$, and moreover that these relationships transfer to (potentially unrelated) sets of futures. We will show how to exploit these relationships to approximately generate samples from a language model $M$ conditioned on some "target history", by only querying $M$ on unrelated sequences. In particular, consider some sequence $h_{\mathrm{targ}} \in \Sigma^{\star}$; typically, to generate continuations $f = (z_1, \ldots, z_m)$ of $h_{\mathrm{targ}}$ under $M$, having gener-

---

[1]Given two $d$-dimensional subspaces $\mathcal{S}, \mathcal{S}' \subset \mathbb{R}^n$, the *principal angles* between $\mathcal{S}, \mathcal{S}'$ are a collection of $d$ real numbers $\theta_1, \ldots, \theta_d \in [0, \pi/2]$ with $\theta_1 \leq \cdots \leq \theta_d$ which represent how much $\mathcal{S}, \mathcal{S}'$ overlap: if $\mathcal{S} = \mathcal{S}'$ then we have $\theta_i = 0$ for all $i$, and if $\mathcal{S}, \mathcal{S}'$ are orthogonal then we have $\theta_i = \pi/2$ for all $i$. See (Golub & Van Loan, 2013, Section 12.4.3) for further discussion.

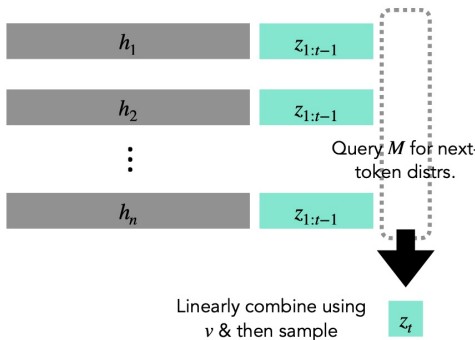

Figure 5: Linear generation (LINGEN) procedure for generating text from a target history using linear combinations of other histories $\mathcal{H} = \{h_1, \ldots, h_n\}$.

ated $z_{1:t-1}$, we query $\Pr_M[\cdot \mid h_{\mathsf{targ}} \circ z_{1:t-1}]$ and sample $z_t$ from it. (For $t = 1$, the sequence $z_{1:0}$ is to be interpreted as the empty sequence.)

We ask: *Can we generate a continuation of $h_{\mathsf{targ}}$ as above without making any queries of the form* $\Pr_M[\cdot \mid h_{\mathsf{targ}} \circ z_{1:t-1}]$? To answer this question, suppose we are given a set of histories $\mathcal{H} \subset \Sigma^\star$, as well as a vector $v = (v_h)_{h \in \mathcal{H}} \in \mathbb{R}^{\mathcal{H}}$. One should interpret $v$ as encoding a way to write $h_{\mathsf{targ}}$ as a linear combination of $h \in \mathcal{H}$ in the sense that for a generic set $\mathcal{F}$ of futures we have $\mathcal{L}_M(h_{\mathsf{targ}}, \mathcal{F}) \approx v^\top \cdot \mathcal{L}_M(\mathcal{H}, \mathcal{F})$. If this holds, then for "typical" sequences $z_{1:t-1}$ we have generated, we can hope that $M$'s next-token distribution, $\mathcal{L}_M(\{h_{\mathsf{targ}}\}, \{z_{1:t-1}\})$, is close to $v^\top \cdot \mathcal{L}_M(\mathcal{H}, \{z_{1:t-1}\})$.

The above intuition suggests the following procedure to generate a sequence of tokens $z_1, z_2, \ldots$. For each $t \geq 1$, having generated $z_{1:t-1}$, for each $h \in \mathcal{H}$, we compute the next-token logit vector $L_{h,t} := L_M[\cdot \mid h \circ z_{1:t-1}] \in \mathbb{R}^\Sigma$ corresponding to $h$. Then we sample the next token $z_t$ from the linear combination of the vectors $L_{h,t}$ as induced by $v$, i.e., $z_t \sim \mathrm{softmax}\left(\sum_{h \in \mathcal{H}} v_h \cdot L_{h,t}\right)$, and repeat for some number $m$ of steps. We call this procedure LIN-GEN; see Figure 5 for an overview, Algorithm 1 (in the appendix) for a complete description, and Section D.4 for a proof that LINGEN generates approximately from $M$'s distribution, under appropriate assumptions paralleling the intuitions discussed above. We instantiate LINGEN in two different manners:

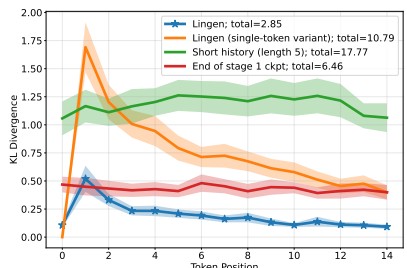

Figure 6: LINGEN with OLMo-1b.

**Option 1: In-distribution target.** In our first experiment, the target $h_{\mathsf{targ}}$ is "in-distribution" for the set of histories $\mathcal{H}$ we use in LINGEN: in particular, the set $\mathcal{H}$ is taken to be subsequences of `wiki` sequences and $h_{\mathsf{targ}}$ is taken to be a subsequence of a `wiki` sequence distinct from all of those producing $\mathcal{H}$. We considered 50 different choices of $h_{\mathsf{targ}}$, and for each generated 5 sequences of length 15 according to LINGEN. The results are shown in Figure 6: we display, at each token index $t$, the KL divergence between the distribution of the next token according to LINGEN (namely, $\mathrm{softmax}\left(\sum_{h \in \mathcal{H}} v_h \cdot L_{h,t}\right)$) and the model $M$ (namely, $\Pr_M[\cdot \mid h \circ z_{1:t-1}]$), averaged over all 250 generations. The "total" numbers reported in the figure represent the average KL divergence, summed over all 15 positions. See Table 2 for some samples generated from LINGEN.

To compute the coefficients $v$ for use by LINGEN, we regressed the rows of $\mathcal{L}_M(\mathcal{H}, \mathcal{F})$ onto $\mathcal{L}_M(\{h_{\mathsf{targ}}\}, \mathcal{F})$. Doing so requires knowledge of $\mathcal{L}_M(h_{\mathsf{targ}}, \mathcal{F})$ and thus requires us to query $\Pr_M[\cdot \mid h_{\mathsf{targ}} \circ f]$ for each $f \in \mathcal{F}$. We expect that, however, by using our observations around model transferability from Section 3.2, such a vector $v$ can be computed instead using some other (perhaps less powerful) model $M'$. We leave further investigation on this point to future work.

In Figure 6, we also display several baselines in which LINGEN is replaced either by (a) in orange, a version of LINGEN where $v$ is only computed by only using, for each history, the *next-token*

*logits* corresponding to the *empty future* (i.e., $\mathcal{L}_M(\mathcal{H}, \{\emptyset\})$), (b) in green, a version of $M$ with restricted context window; or (c) in red, an earlier training checkpoint of $M$. We emphasize that (a) performs well at generating the first token $z_1$ (as expected), but performs significantly worse at later tokens $z_t$, $t > 1$: this point emphasizes **the importance of considering the extended logit matrix $\mathcal{L}_M(\mathcal{H}, \mathcal{F})$ as opposed to the single-token logit matrix** which has been considered in prior works (Yang et al., 2017). Altogether, LINGEN performs significantly better than all baselines. [2]

**Option 2: Out-of-distribution target (& futures).** Next, we consider a more challenging setting where the target $h_{\text{targ}}$ is unrelated to the set of histories $\mathcal{H}$ used in LINGEN. We took the same 50 targets $h_{\text{targ}}$ (origining from `wiki`) as described above, but now let the set of histories $\mathcal{H}$ be the set $\mathcal{H}^{\text{nonsense}}$ obtained by randomly permuting all tokens amongst all elements of $\mathcal{H}$. Moreover, when computing $v$, we replace the set of futures $\mathcal{F}$ with $\mathcal{F}^{\text{nonsense}}$ (i.e., obtained by randomly permuting all tokens). The results are shown in Figure 1b; while the KL divergences corresponding to LINGEN are generally larger than in Figure 6, they are still smaller than all baselines, and the generated sequences (Table 3) generally show clear evidence of incorporating information from $h_{\text{targ}}$. Thus, by using the low-rank structure of the logit matrix, **we can generate starting from a prompt by only querying the model at (nonsense) sequences unrelated to the prompt.** While we leave a more thorough investigation for future work, we believe that this approach suggests novel ways to create *jailbreaks* for LLMs, an area that has seen extensive work recently (Wei et al., 2023; Yi et al., 2024): for instance, querying the model on sequences in $\mathcal{H}^{\text{nonsense}}$ as above may allow us to circumvent input filters which aim to filter out harmful prompts; see Section E for further discussion. In addition to implications for AI safety, we stress that the structure we observe is fundamental and could lead to many other applications ranging from computational efficiency at inference time to improved training procedures. We discuss this and other future directions in Section E.

## 4 THEORETICAL RESULTS

In the following sections, we will develop theoretical foundations for understanding low logit rank.

**Definition 4.1** (Logit Rank). *A language model $M$ has* logit rank $d$ *if for any set of histories $\mathcal{H} \subset \Sigma^*$ and futures $\mathcal{F} \subset \Sigma^*$, the matrix $\mathcal{L}_M(\mathcal{H}, \mathcal{F})$ has rank at most $d$.*

We will show that low logit rank is equivalent to a simple generative model and then explore the representation power of this model. Finally, we will show that given query access to a language model with low logit rank, we can efficiently learn a description of a language model that approximates it well in TV distance. These results taken together establish low logit rank as a natural and tractable theoretical model for understanding language models.

### 4.1 A LOW RANK GENERATIVE MODEL

Our first main result is demonstrating that the low logit rank condition can be captured by a simple generative model. The idea is to consider a version of a linear dynamical system model but with two important differences: (1) the dynamics are allowed to depend on the current sampled token and the current timestep and (2) the observations are generated through a softmax nonlinearity. This model is an extension of the Input Switched Affine Network (ISAN) model studied by Foerster et al. (2017) as an interpretable recurrent architecture (see Theorem D.1), with the difference being that our model allows the dynamics to change every timestep. We call our model a time-varying ISAN.

**Definition 4.2** (Time-varying ISAN). *A time-varying ISAN of sequence length $T$ is specified by matrices $A_{z,t} \in \mathbb{R}^{d \times d}, B_t \in \mathbb{R}^{\Sigma \times d}$ for $z \in \Sigma, t \in [T]$ and an initial state $x_0 \in \mathbb{R}^d$. It defines a distribution over sequences of tokens $z_1, z_2, \ldots, z_T$ by: $z_t$ is sampled from $\text{softmax}(B_t x_{t-1})$. Then, the hidden state is updated as $x_t = A_{z_t,t} x_{t-1}$ [3]. We refer to $d$ as the hidden dimension [4].*

---

[2] In general, LINGEN improves at later tokens; we believe this holds since for such tokens, the distribution of the next token depends more on the "more recent" previously generated tokens $z_{1:t-1}$ (as opposed to $h_{\text{targ}}$), and these tokens *are* fed into the model to compute $L_{h,t}$. One exception is that LINGEN performs very well at the first token, which may be since the single-token logit matrix of LLMs *is* low-rank (see Section 1).

[3] Note that incorporating a bias term, i.e. $x_t = A_{z_t,t} x_{t-1} + b_{z_t,t}$, is equivalent to adding an extra dimension that is always 1. Thus, we will use the above form without the bias term.

[4] For simplicity, we will not have an end-of-sequence token, but will treat the time-varying ISAN language model as defining a distribution over sequences of a fixed length.

The (time-varying) ISAN captures the linear compression perspective since the hidden state $x_t$ can be seen as a compression of the history $\{z_i\}_{i=1}^t$ that is sufficient to generate the future $\{z_j\}_{j=t+1}^T$. This definition also exhibits a general form of relational linearity (see e.g. Marconato et al. (2024); Hernandez et al. (2023); Paccanaro & Hinton (2002)) — any sequence of tokens, not just those expressing relations in a typical semantic sense, can be viewed as a linear operation applied to the embedding (the hidden state) of the history before it.

As mentioned, a similar time-invariant model has been studied by Foerster et al. (2017) for its interpretability, but our perspective differs because rather than viewing it as a separate architecture, we treat it as a generative model that can approximate modern language models. In fact, the work by Foerster et al. (2017) supports the usefulness of this model as a theoretically tractable surrogate, as although far from state of the art, it can achieve relatively coherent language modeling at small scales. We present a new motivation for studying this model by proving that the extended logit matrix being low rank is equivalent to a language model being expressible as a time-varying ISAN.

**Theorem 4.3** (Equivalence between Low Logit Rank and Time-Varying ISAN). *Let $M$ be a language model over sequences of length $T$. Then $M$ is expressible as a time-varying ISAN with hidden dimension $d$ if and only if the logit matrix $\mathcal{L}_M(\Sigma^t, \Sigma^{\leq T-t})$ has rank at most $d$ for all $t \leq T$.*

## 4.2 Representation Power

We show that ISANs can represent a variety of simple architectures and languages. Specifically, we show that they can represent linear SSM layers, and can express copying and noisy parity. The formal statements and proofs are deferred to Section D.2.

## 4.3 Provable Learning Guarantees

Given that ISANs can express noisy parity, and the standard cryptographic assumption on the hardness of learning a noisy parity, *it is computationally hard to learn an ISAN from samples* (see Theorem D.13). To circumvent this, we consider a *logit query* model, where the learner can query the logits for the next token given any history. This model mirrors settings for practical model stealing attacks e.g. (Carlini et al., 2024), for stealing the embedding dimension and output embedding matrix for production language models and also connects to classical and modern works on computational learning theory (Angluin, 1987; Mahajan et al., 2023; Liu & Moitra, 2025).

Our main result is the following theorem, which shows that given logit query access to a time-varying ISAN, we can efficiently learn a time-varying ISAN that approximates it well in TV distance.

**Theorem 4.4.** *Given logit query access to an unknown time-varying ISAN $M$ over $\Sigma^T$ with hidden dimension at most $d$, there is an algorithm that uses $\mathrm{poly}(d, |\Sigma|, T, 1/\epsilon)$ runtime and queries and returns a description of a time-varying ISAN $M'$ such that $\mathbb{E}[D_{\mathrm{TV}}(M, M')] \leq \epsilon$.*

## 5 Conclusions

Our paper develops a simple but mathematically justified framework built on the low-rank structure of the extended logit matrix and demonstrates its empirical predictive power across different models, data, and tokenizers. We believe this low-rank generative viewpoint will provide a general model-agnostic foundation for understanding, probing, and steering language models. While some of the empirical demonstrations in this paper such as LINGEN are proof-of-concept, we believe this lens points to many promising future directions, both for new understandings and practical applications, which we discuss in more detail in Section E.

## Acknowledgements

AL was supported by a Miller Research Fellowship. We are grateful to Adam Block, Akshay Krishnamurthy, and Ankur Moitra for helpful comments.

ETHICS STATEMENT

We view the results in this paper as fundamental research and do not see ethical concerns. While some of the insights may eventually lead to attacks or jailbreaks against LLMs, discovering such attacks is an important part of the research process, and we believe that the benefits outweigh the concerns and that the insights in this paper will help towards better defenses as well.

REPRODUCIBILITY STATEMENT

Full experimental specifications are described in detail in Section B, and full proofs of our theoretical results are given in Section D. We plan to make the code for our experiments available.

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

## A    RELATED WORK

**Low Rank Structure in Language Models**    There has been significant work understanding the implications of the low rank structure in the next-token logit matrix. In particular, the output embedding matrix constrains the vector of next-token logits to be in a space of dimension equal to the hidden dimension (rather than the vocabulary size) and thus limits the family of conditional distributions that can be realized. This phenomenon, often called the *softmax bottleneck*, was formalized in Yang et al. (2017). The implications of this for language modeling are also studied in Press & Wolf (2016); Kanai et al. (2018); Finlayson et al. (2023). Furthermore, recent works study this softmax bottleneck from a different perspective, showing that it leaks information about models even behind black-box APIs and results in vulnerabilities to model stealing attacks (Finlayson et al., 2024; Carlini et al., 2024). However, these works all focus on just the next-token logit matrix, whereas our framework allows us to reason about generating much longer sequences.

There have also been many works that observe low-rank structure in the weight matrices of machine learning models, dating back to the early days of deep learning (Denil et al., 2013), and this has been used to understand (Aghajanyan et al., 2020) and develop efficient methods for fine-tuning large models e.g. LoRA and its variants (Zhang et al., 2023; Hu et al., 2022). While similar in philosophy, our approach is very different because it is model-agnostic, does not involve looking at the weights, but instead finds low-rank structure directly in the logits over sequences of tokens.

Linear structure in word embeddings has been observed in well-known works (Mikolov et al., 2013; Goldberg & Levy, 2014; Levy & Goldberg, 2014) and also to some extent for sentence embeddings (Zhu & De Melo, 2020; Bowman et al., 2016; Li et al., 2020). The linear representation hypothesis is, informally, the idea that concepts are represented linearly as directions in some representation space. Park et al. (2023) propose a formalization of this notion for language models, showing that certain concepts for the next token, such as gender, can be represented linearly in either the logit vector or the last hidden state. There have also been many works, that broadly fall under mechanistic-interpretability, on probing and steering using the linear representations in internal layers of the model e.g. Elhage et al. (2021); Meng et al. (2022); Hernandez et al. (2023); Nanda et al. (2023); Turner et al. (2023); Todd et al. (2023); Hendel et al. (2023); Li et al. (2023); Geva et al. (2022). Compared to these aforementioned works, our framework only uses the output logits, but also allows us to reason about generating longer sequences of tokens (beyond just the next token). It is an exciting direction to try to extract features and perform interventions through the representations provided by our framework.

**Related Theoretical Models**   Linear dynamical systems form the basis of control theory and are also the key building block in modern state-space models. Given the vast literature in control theory, we refer the reader to (Kailath, 1980; Chen, 1999) for a classical treatment and (Hazan & Singh, 2025) for a modern treatment. A sequence of works over the past few years (Gu et al., 2021a; Gu & Dao, 2023; Katharopoulos et al., 2020; Dao & Gu, 2024; Gupta et al., 2022; Gu et al., 2021b) has led to the development of modern state-space models, introducing additional twists such as selective gating (which is related to the token-dependent transitions we use) and connections to linearized transformers. Further connections to linear dynamical systems have led to new theoretically motivated architectures (Agarwal et al., 2024). We believe our work presents a new perspective that is both theoretically and empirically justified, while maintaining the conceptual simplicity of linear dynamical systems.

In terms of simpler, mathematically tractable generative models, the ISAN model is related to weighted finite automata (Droste et al., 2009) and hidden Markov models (HMMs) (Rabiner & Juang, 2003), except with the addition of a softmax nonlinearity to sample the next token. This nonlinearity significantly improves its empirical performance on language modeling compared to these earlier models (Foerster et al., 2017). There have been many earlier works studying the learnability of these classical models such as Mossel & Roch (2005); Hsu et al. (2012); Balle et al. (2014). Query learning settings have been classically studied in the literature on learning automata and formal languages (Angluin, 1987), and more recently received renewed interest in the context of model stealing for large language models (Mahajan et al., 2023; Liu & Moitra, 2025). Compared to these works which study classical models (e.g. HMMs), we believe the theoretical work here takes another step closer to modern language models.

# B   EXPERIMENTAL DETAILS

## B.1   ADDITIONAL EXPERIMENTAL DETAILS FROM SECTION 3.1

### B.1.1   SETUP FOR EXPERIMENTS

**Choice of datasets and models.**   We considered the following choices for the autoregressive language model $M$ and the dataset $D$:

1. The model $M$ was chosen amongst OLMo-1b, Olmo-7b (OLMo et al., 2024), Llama-1b (et al., 2024), Mamba-1.4b (Gu & Dao, 2024), Gemma-1b (et al., 2025). Moreover, as a baseline comparison, we considered the checkpoint for OLMo-1b at time step $0$ of training.

2. The dataset $D$ was chosen amongst: the `wiki`, `arxiv`, `starcoder` subsets of the `olmo-mix-1124` (OLMo et al., 2024) training dataset; the `math` subset of the `dolmino-mix-1124` dataset (OLMo et al., 2024); and the `c4` dataset (Raffel et al., 2020).

Next, we detail the construction of the sets $\mathcal{H}, \mathcal{F}$ from a dataset $D$. The datasets $D$ we used each consisted of a collection of many sequences, i.e., we have $D = \{y^{(1)}, \ldots, y^{(N)}\}$ for some large $N$.

**Construction of histories.** To generate a set $\mathcal{H}$ of histories of size $n$, we fixed parameters $\ell_{\min}, \ell_{\max}$. We first chose a random subset $\mathcal{S}$ of $n$ elements of $D$ of length at least $\ell_{\max}$ (as measured by the number of characters).[5] For each sequence $y \in \mathcal{S}$, we selected a uniformly random contiguous subsequence of $y$ of length $\ell_{\max}$. We then chose $\ell \sim \mathrm{Unif}[\ell_{\min}, \ell_{\max}]$, and added sequence $y_{1:\ell}$ to $\mathcal{H}$ (rounded to the nearest full word). We opted to measure length (and perform truncation) with respect to characters (as opposed to tokens) since we wish to use the same set of histories (and futures) for different models, which typically have different tokenizers.

**Construction of futures.** The set of futures $\mathcal{F}$ was generated identically to the set $\mathcal{H}$ of histories, with the exception that in the context of the previous paragraph, $y_{\ell:|y|}$ (where $|y|$ denotes the length of $y$) was added to $\mathcal{F}$.

**Computation of the exponent $\alpha$.** To compute the power law exponents $\alpha$ shown in Figures 2 and 7, we used least-squares regression, as follows. Given a matrix $L \in \mathbb{R}^{n \times m}$ (where $n \leq m$), we let its singular values, arranged in decreasing order (excluding the largest) be $\sigma_1, \ldots, \sigma_{n-1}$. We performed a 2-dimensional linear regression with covariates $(\log(i/n), \log((n-i)/n))$ and labels $\log \sigma_i$, for $i \in [n-1]$. Adding the covariate $\log((n-i)/n)$ yielded a better fit as it accounts for the precipitous drop in $\sigma_i$ as $i$ approaches the number of rows $n$ (when the power law as described in Section 3.1 ceases to describe the decay of the singular values $\sigma_i$). The resulting coefficient for the $\log((n-i)/n)$ covariate was always very small, meaning that its effect was negligible for $i \ll n$.

Moreover, in the regression, the data point corresponding to index $i$ was weighted by $1/i$. We chose this weighting (as opposed to, say, uniformly weighting all data points) to account for the logarithmic scale of the covariates in terms of their dependence on $i$.

### B.1.2 Details for singular value plots

Recall that for an integer $k$, $\mathcal{L}_{M,k}(\mathcal{H}, \mathcal{F})$ denotes the sub-matrix obtained from $\mathcal{L}_M(\mathcal{H}, \mathcal{F})$ by selecting, for each $f \in \mathcal{F}$, the columns $(f, z)$ indexed by tokens $z$ which are one of the $k$ most likely next tokens following $f$, with respect to $M$. Figures 2 and 7 show the singular values for logit matrices $\mathcal{L}_{M,50}(\mathcal{H}, \mathcal{F})$ for fixed sets $\mathcal{H}, \mathcal{F}$ of histories and futures of size approximately 10000,[6] as generated according to the procedure from Section B.1.1. For this procedure we chose $\ell_{\min} = 50, \ell_{\max} = 200$. The figures also show the singular values for $\mathcal{L}_{M,50}(\mathcal{H}_i, \mathcal{F}_i)$, where $\mathcal{H}_i$ (resp., $\mathcal{F}_i$) contains the first $|\mathcal{H}|/\sqrt{i}$ (resp., $|\mathcal{F}|/\sqrt{i}$) elements of $\mathcal{H}$ (resp., $\mathcal{F}$), for $i \in \{2, 4, 8, 16\}$. Thus the number of elements of $\mathcal{L}_{M,50}(\mathcal{H}_i, \mathcal{F}_i)$ is roughly $1/i$ that of $\mathcal{L}_{M,50}(\mathcal{H}, \mathcal{F})$. Singular values were normalized by the square root of the number of elements of each matrix.

Next, Figure 9 shows the singular values of $\mathcal{L}_{M,50}(\mathcal{H}, \mathcal{F})$ when $M$ was OLMo-1b and $\mathcal{H}, \mathcal{F}$ were obtained via the procedure described in Section B.1.1 when the dataset $D$ was taken to be `arxiv, starcoder, math, c4`. We remark that for these plots the sets $\mathcal{H}, \mathcal{F}$ were taken to be of size only 4000.

### B.1.3 Details for KL divergence plots

Figures 1a, 8 and 10 show the average KL errors between the logit matrices considered in Figures 2, 7 and 9, respectively, to low-rank approximations. Low-rank approximations were computed using an approximate singular value decomposition (via `torch.svd_lowrank`) and then zeroing out all singular values apart from the largest $r$ ones. (Note that, for an exact SVD, this procedure gives the closest rank-$r$ approximation in Frobenius norm.) Since we are considering the submatrices $\mathcal{L}_{M,50}(\mathcal{H}, \mathcal{F})$, these KL divergences are taken with respect to distributions restricted to the 50 most likely tokens per future.

The dashed line at the top of the figures represents the following rank-1 baseline: for each future $f$, we simply use the next-token distribution for $f$, i.e., $\mathrm{Pr}_M[\cdot \mid f]$, to approximate all entries (i.e., rows) corresponding to $f$. We remark that this approximation is in general not the optimal rank-1 approximation to the logit matrix (with respect to Frobenius norm).

---

[5]For some of the larger datasets, we used a buffer size of at least $10^6$ to stream the dataset, so technically the set of histories was not a uniformly random subset.

[6]In particular, we generated sets $\mathcal{H}, \mathcal{F}$ of size exactly 10000 and then filtered out 0-length futures, of which there were approximately 60.

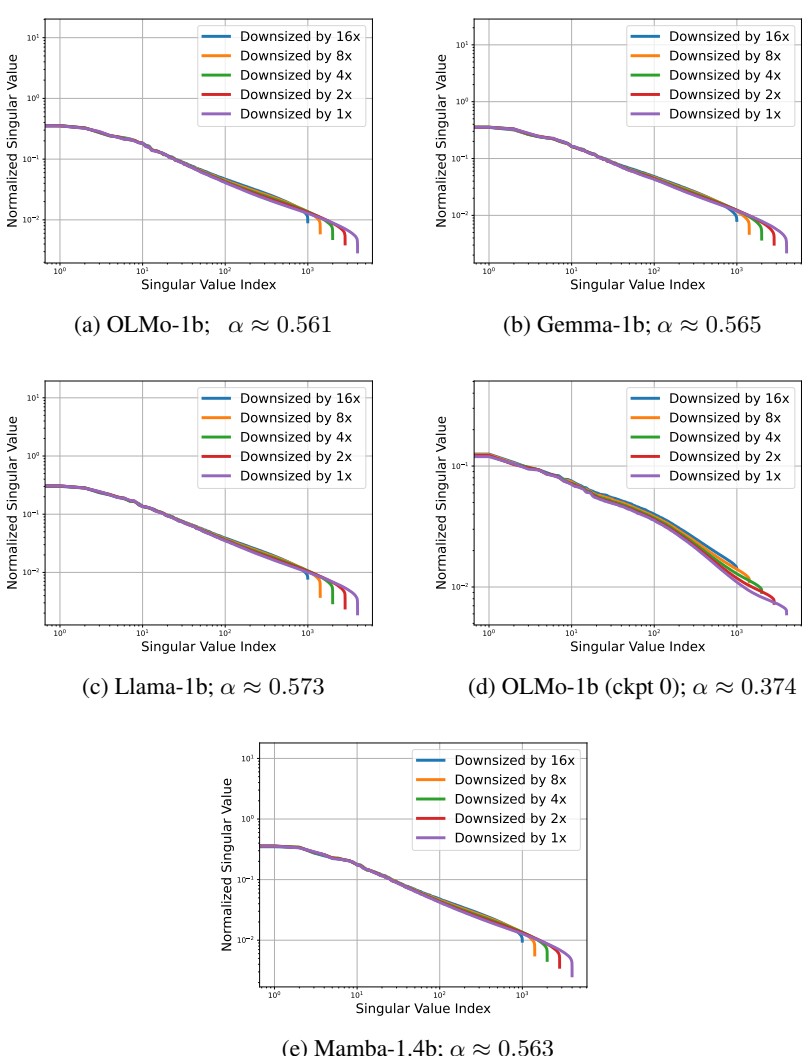

(a) OLMo-1b; $\alpha \approx 0.561$

(b) Gemma-1b; $\alpha \approx 0.565$

(c) Llama-1b; $\alpha \approx 0.573$

(d) OLMo-1b (ckpt 0); $\alpha \approx 0.374$

(e) Mamba-1.4b; $\alpha \approx 0.563$

Figure 7: Singular values for $\mathcal{L}_{M,50}(\mathcal{H}, \mathcal{F})$ for various models $M$; see Section B.1.2.

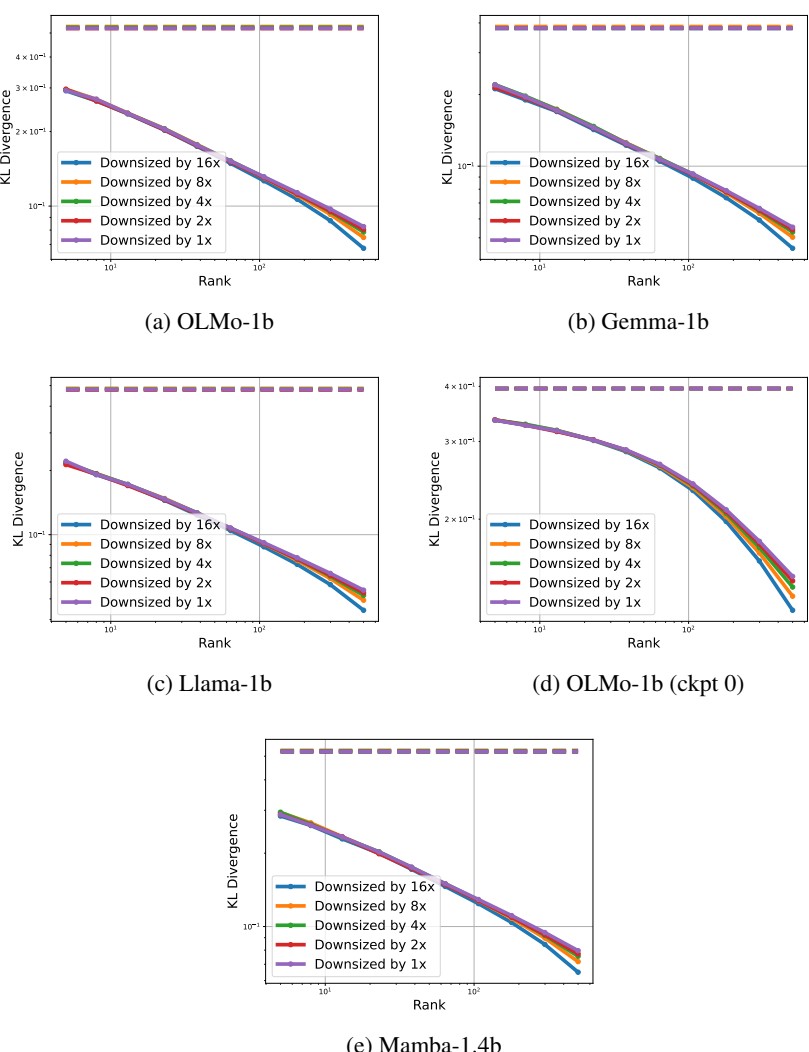

Figure 8: Average KL divergence to a low-rank approximation for various models $M$; see Section B.1.3.

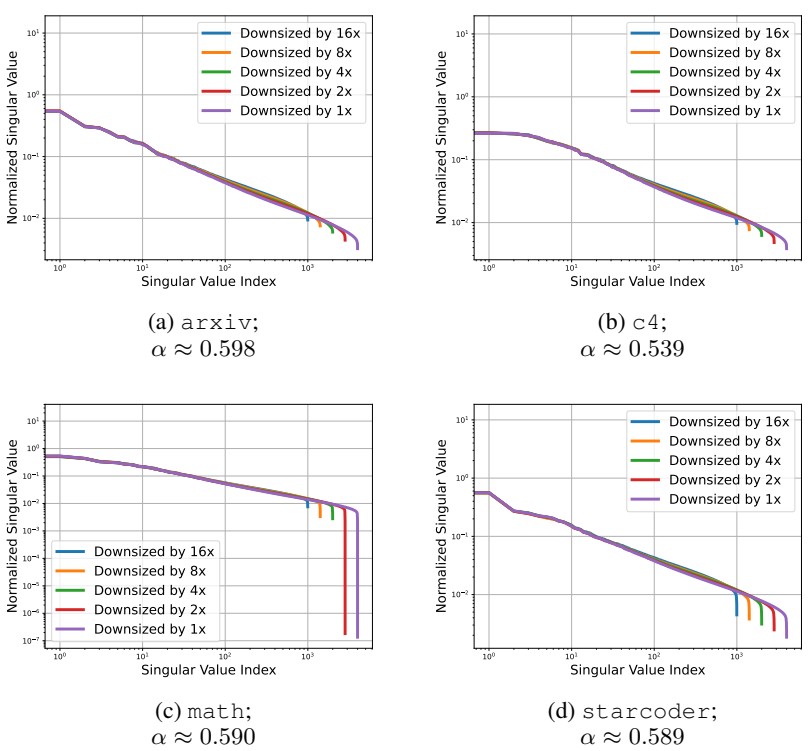

Figure 9: Singular values of the logit matrix $\mathcal{F}_{M,50}(\mathcal{H}, \mathcal{F})$ when $\mathcal{H}, \mathcal{F}$ are chosen from various datasets as detailed in Section B.1.1. The model $M$ was OLMo-1b.

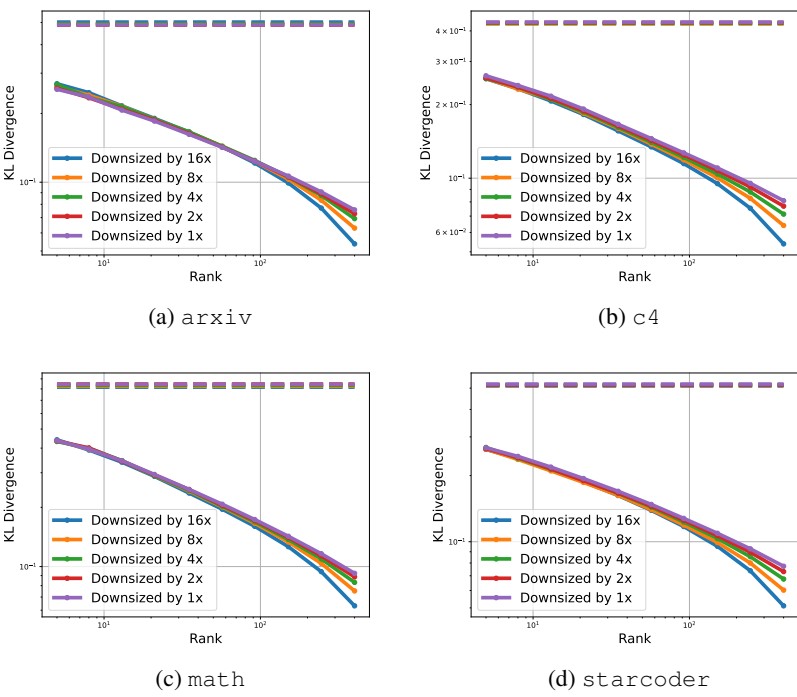

Figure 10: Average KL errors to a low-rank approximation, for the same logit matrices as described in Figure 9.

### B.1.4 Supporting theoretical results

Below we collect a couple of standard facts which we use in our discussion of the experiments. For completeness, we provide their proofs (which are standard).

**Fact B.1** (Phase transition in singular value decay). *Suppose $L \in \mathbb{R}^{n \times m}$ with $n \leq m$, and its singular values $\sigma_1, \ldots, \sigma_n$ satisfy $\sigma_i \asymp C \cdot i^{-\alpha}$, for some constants $C, \alpha > 0$. Then:*

1. *If $\alpha > 1/2$, then for any $r \in \mathbb{N}$, there is a matrix $A$ of rank $r$ for which $\|L - A\|_F \leq \|L\|_F \cdot O(r^{\frac{1}{2} - \alpha})$. (Here $\| \cdot \|_F$ represents the Frobenius norm.)*

2. *If $\alpha < 1/2$, then any matrix $A$ of rank $r \leq \frac{n}{2^{1/(1-2\alpha)}}$ satisfies $\|L - A\|_F \geq \Omega(\|L\|_F)$.*

*In the above statements, the $O(\cdot), \Omega(\cdot)$ hide absolute constants.*

*Proof.* The fact is standard, but we provide the proof anyways. Let us suppose that $C_0, C_1 > 0$ are so that $C_0 \cdot i^{-\alpha} \leq \sigma_i \leq C_1 \cdot i^{-\alpha}$ for all $i$.

**Case 1:** $\alpha > 1/2$. First suppose that $\alpha > 1/2$. We may write the SVD of $L$ as $\sum_{i=1}^n \sigma_i \cdot u_i v_i^\top$, where $\{u_i\}$ and $\{v_i\}$ are orthonormal bases of $\mathbb{R}^n$ and $\mathbb{R}^m$, respectively. Note that $\|L\|_F = \sqrt{\sum_{i=1}^n \sigma_i^2} \geq C_0 \cdot \Omega\left(\sqrt{\frac{1}{2\alpha - 1}}\right)$.

Fix any $r \in \mathbb{N}$, and let $A = \sum_{i=1}^r \sigma_i u_i v_i^\top$. Then $\|L - A\|_F = \sqrt{\sum_{i=r+1}^n \sigma_i^2} \leq C_1 \cdot O\left(\sqrt{\frac{r^{1-2\alpha}}{2\alpha - 1}}\right)$. It follows that

$$\frac{\|L - A\|_F}{\|L\|_F} \leq O\left(\frac{C_1}{C_0} \cdot r^{\frac{1}{2} - \alpha}\right),$$

where the $O(\cdot)$ hides dependence on an absolute constant, as desired.

**Case 2:** $\alpha < 1/2$. It is a standard fact that for any $r$, the best rank-$r$ approximation to a matrix $L$ is given by truncating its SVD at rank-$r$, i.e., for any rank-$r$ matrix $A$, we have that

$$\|L - A\|_F \geq \left\| L - \sum_{i=1}^r \sigma_i u_i v_i^\top \right\|_F = \sqrt{\sum_{i=r+1}^n \sigma_i^2} \geq C_0 \cdot \Omega\left(\sqrt{\frac{n^{1-2\alpha} - r^{1-2\alpha}}{2\alpha - 1}}\right).$$

But $\|L\|_F \leq C_1 \cdot O\left(\sqrt{\frac{n^{1-2\alpha}}{1-2\alpha}}\right)$, meaning that for $r \leq \frac{n}{2^{1/(1-2\alpha)}}$, we have $\frac{\|L-A\|_F}{\|L\|_F} \geq \Omega(C_0/C_1)$, as desired. $\square$

**Fact B.2.** *Consider sets $\mathcal{H}, \mathcal{F}$ of sizes $n, m$ respectively. Then for any matrices $A, A' \in \mathbb{R}^{\mathcal{H} \times (\mathcal{F} \times \Sigma)}$, it holds that*

$$\sum_{h \in \mathcal{H}, f \in \mathcal{F}} D_{\mathsf{KL}}(\mathrm{softmax}(A_{h,f}) \| \mathrm{softmax}(A'_{h,f})) \leq \frac{1}{2} \|A - A'\|_F^2.$$

*Proof.* It follows immediately from the definition of average KL divergence that it suffices to prove the following fact: if $v, v' \in \mathbb{R}^d$, then letting $p = \mathrm{softmax}(v)$, $p' = \mathrm{softmax}(v')$ (so that $p, p' \in \Delta^d$), then we have $D_{\mathsf{KL}}(p \| p') \leq \|v - v'\|_2^2$. To prove this fact, let us define $F(v) = \log \sum_{i=1}^d e^{v_i}$. Then a direct computation yields

$$D_{\mathsf{KL}}(p \| p') = F(v) - F(v') - \langle \nabla F(v), v' - v \rangle. \tag{1}$$

Next, for any $x \in \mathbb{R}^d$, for $q = \mathrm{softmax}(x)$ we have $\nabla^2 F(x) = \mathrm{diag}(q) - qq^\top \preceq I_d$, i.e., $F$ is 1-gradient Lipschitz. But this yields that the right-hand side of Equation (1) is bounded above by $\frac{1}{2} \cdot \|v - v'\|_2^2$, as desired. $\square$

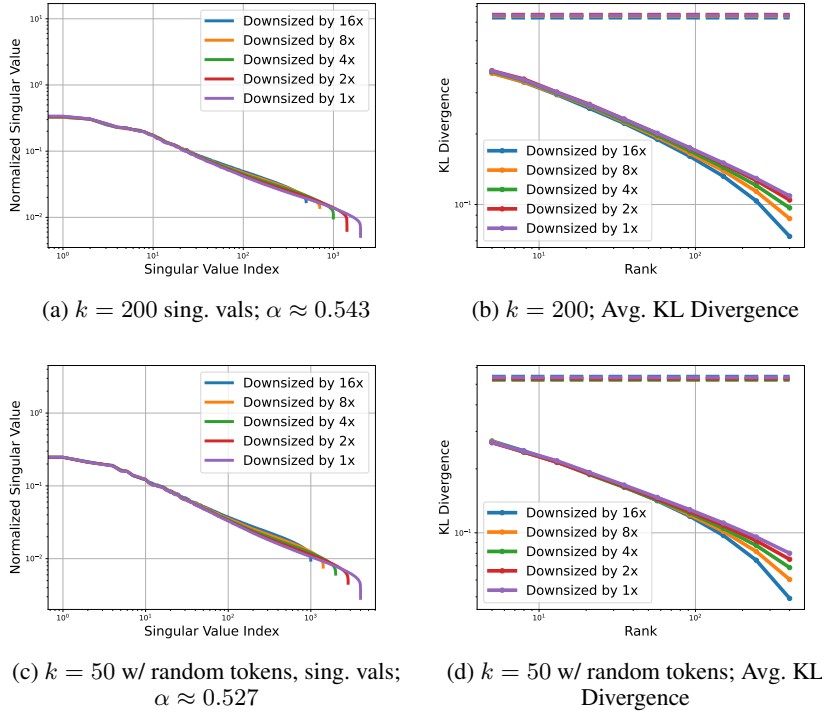

(a) $k = 200$ sing. vals; $\alpha \approx 0.543$      (b) $k = 200$; Avg. KL Divergence

(c) $k = 50$ w/ random tokens, sing. vals;      (d) $k = 50$ w/ random tokens; Avg. KL
$\alpha \approx 0.527$      Divergence

Figure 11: (**(a)**, **(b)**): Low-rank approximability of $\mathcal{L}_{M,200}(\mathcal{H}, \mathcal{H})$. **(c)**, **(d)**: Low-rank approximability of $\tilde{\mathcal{L}}_{M,50}(\mathcal{H}, \mathcal{F})$; see Section B.1.5.

### B.1.5 ABLATIONS FOR THE PARAMETER $k$

Recall that our experiments exhibiting approximate low-rank structure of the logit matrix actually apply to the sub-matrices $\mathcal{L}_{M,k}(\mathcal{H}, \mathcal{F})$ described in Section 3.1, for $k = 50$. In Figure 11, we present two modifications to this approach (all with OLMo-1b): in Figures 11a and 11b we measure the low-rank approximability of $\mathcal{L}_{M,200}(\mathcal{H}, \mathcal{F})$, for sets $\mathcal{H}, \mathcal{F}$. Due to memory limitations, we took $|\mathcal{H}|, |\mathcal{F}|$ to be of size approximately 2000 in Figure 11a and to be of size approximately 5000 for Figure 11b.

Next, in Figures 11c and 11d, we measure the low-rank approximability of a submatrix $\tilde{\mathcal{L}}_{M,50}(\mathcal{H}, \mathcal{F})$ obtained from $\mathcal{L}_M(\mathcal{H}, \mathcal{F})$ by sampling *randomly* 50 tokens per future and selecting the corresponding columns of $\mathcal{L}_M(\mathcal{H}, \mathcal{F})$. For these figures we used $\mathcal{H}, \mathcal{F}$ of sizes approximately 4000.

All of the aforementioned ablations show similar evidence of low-rank structure similarly to the logit matrices $\mathcal{L}_{M,50}(\mathcal{H}, \mathcal{F})$. We remark that, in general, as the size of $\mathcal{H}, \mathcal{F}$ are increased, the fitted power law exponent $\alpha$ describing the decay of the singular values tends to decrease slightly, perhaps because the effect of the slight "hump" for the first few singular values is diminished. This may explain why the exponents $\alpha$ in Figures 11a and 11c are slightly smaller than in Figure 7a.

### B.2 SUBSPACE COMPARISON

Figure 4 displays the principal angles between low-rank approximations of $\mathcal{L}_{M,50}(\mathcal{H}, \mathcal{F})$ and $\mathcal{L}_{M,50}(\mathcal{H}, \mathcal{F}^{\text{nonsense}})$ where the model $M$ was OLMo-1b, $\mathcal{H}, \mathcal{F}$ were generated from `wiki` as described in Section B.1.1, with $|\mathcal{H}| = 10000$ and where $\mathcal{F}, \mathcal{F}^{\text{nonsense}}$ were each of size approximately 5000.[7] Low-rank approximations were computed by truncating (approximate) singular value decompositions, as described in Section B.1.3. The dashed lines show the principal angles between a pair of independent uniformly random subspaces of the indicated dimension. It is evident that the

---

[7]We started with sets $\mathcal{F}, \mathcal{F}^{\text{nonsense}}$ of size 5000 and then removed a small number of 0-length futures.

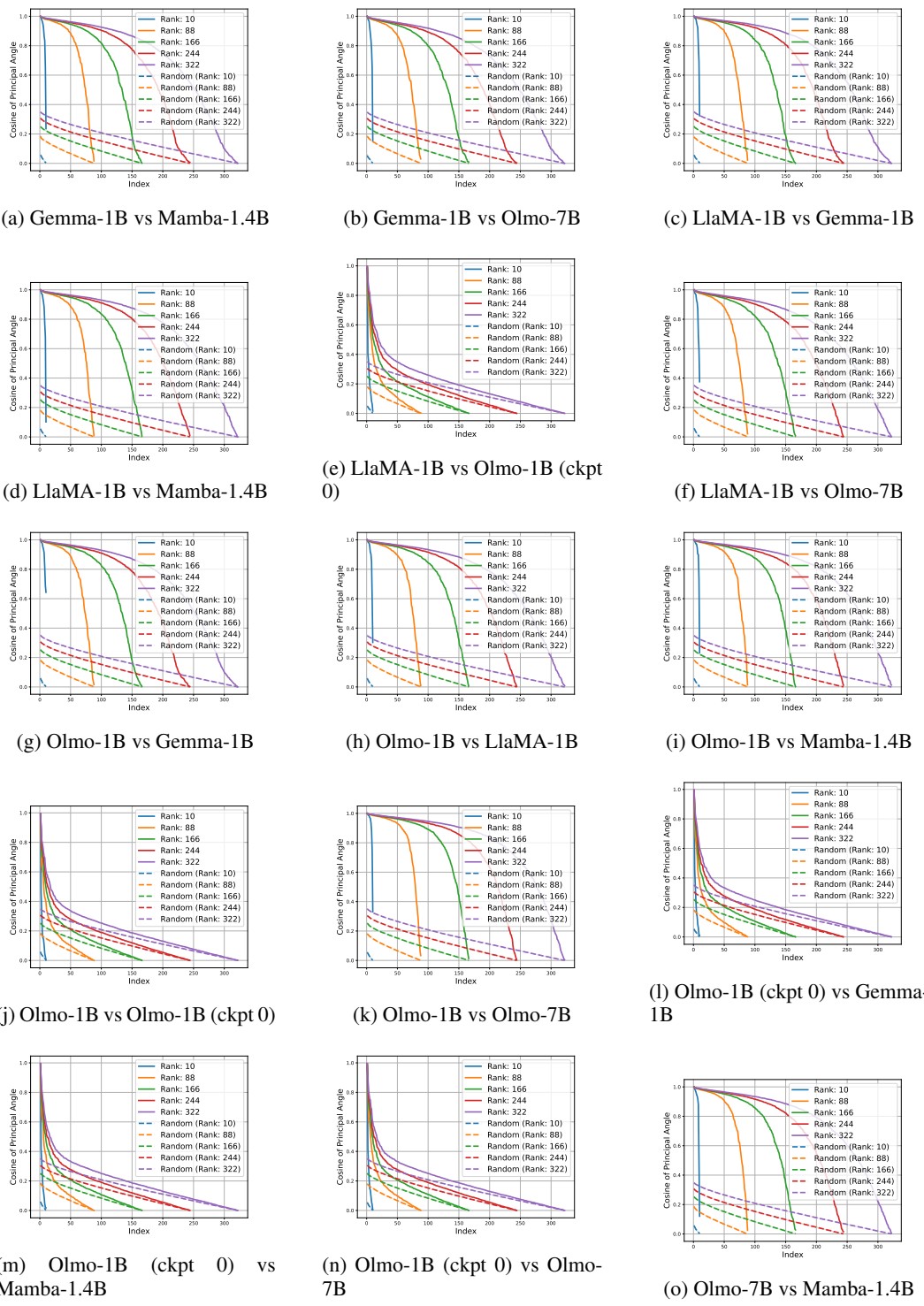

(a) Gemma-1B vs Mamba-1.4B

(b) Gemma-1B vs Olmo-7B

(c) LlaMA-1B vs Gemma-1B

(d) LlaMA-1B vs Mamba-1.4B

(e) LlaMA-1B vs Olmo-1B (ckpt 0)

(f) LlaMA-1B vs Olmo-7B

(g) Olmo-1B vs Gemma-1B

(h) Olmo-1B vs LlaMA-1B

(i) Olmo-1B vs Mamba-1.4B

(j) Olmo-1B vs Olmo-1B (ckpt 0)

(k) Olmo-1B vs Olmo-7B

(l) Olmo-1B (ckpt 0) vs Gemma-1B

(m) Olmo-1B (ckpt 0) vs Mamba-1.4B

(n) Olmo-1B (ckpt 0) vs Olmo-7B

(o) Olmo-7B vs Mamba-1.4B

Figure 12: Principal angles between column spaces of logit matrices across all pairs of models; see Section B.2.

| History | Closest history |
|---|---|
| Investigations personnel in 1927. The name, derived from the eighth letter of the Greek alphabet, appears to have been first used on a 1946 Argentine government chart following surveys | (BIT) was created on 9 December 1898, in response to the Bombay plague epidemic of 1896. It was created |
| on Honest Jon's capitalised on this with Theo Parrish, Rashad & Spinn, Ricardo | for Nelly Furtado, N-Dubz, Nina Sky, Lil' Flip, Noel Gourdin, Starboy Nathan, Shawnna, Jimmy Cozier, Alison Hinds, Jazmine Sullivan, I 20, Cory Lee, Chris Webby, City |
| Shiroro Airfield is an airstrip serving the village of Shiroro and the Shiroro Hydroelectric Power Station in the Niger State of Nigeria. The runway is | Canadian Forces Base located immediately south of the town of Chatham, New Brunswick, Canada. Parts are now operating as Miramichi Municipal Airport since 1974 with a partial runway |
| to numerous publications, and supported the development of a range of new PES schemes including the BioCarbon Fund at the World Bank and the Mexican PES | as a mitigation bank providing ecosystem services to the public in the form of Environmental |
| Thereupon, Gangaram slays out Janardhan Seth, perceiving the dirty deed angered Rambabu reaches Janardhan Seth's | stealing the idol's jewellery and his wife is accused of being a witch. They are killed by a mysterious person after which Chandni's father Yash Narayana Vashisht becomes the Mahant. Dev escapes |
| process of removing government-controlled entry and price restrictions on airlines affecting, in particular, the carriers | stated model. It maintains close ties to other Green parties |
| On Spitfires. Scotland was in range of Nazi Germany's long-range bombers and reconnaissance aircraft. The Luftwaffe's | minor improvements that increased their anti-aircraft capabilities. Their crew numbered 228 officers and enlisted men. The |
| U-boat built for the Nazi Germany's "Kriegsmarine" for service during World War II. | floatplane of the 1910s produced by Flugzeugbau Friedrichshafen. |
| European works, classical Chinese works and Nôm works into Quoc Ngu - modern | supported the creation of the Empire of China and the 1917 Manchu |
| appointed Assistant Chaplain at Geelong Grammar School from 1959 to 1961, vicar of Romsey and Sunbury with Lancefield from 1961 to 1964 | the Young Women's Christian Association of New Zealand. She was a member of its Dunedin board from 1930 until 1944, Dunedin |

Table 1: For 10 values of $h \in \mathcal{H}$ (left column) we display the history $h' \in \mathcal{H}, h' \neq h$ minimizing the distance between the respective rows of the logits matrix, i.e., $\|\mathcal{L}_M(\{h\}, \mathcal{F}) - \mathcal{L}_M(\{h'\}, \mathcal{F})\|_2$. Examples were not cherry-picked (they correspond to the first 10 examples in the set $\mathcal{H}$ described in Section 3.1, as obtained from wiki).

column spaces of the logit matrices have significantly more overlap than what would be expected from random subspaces.

In a similar manner, Figure 12 shows the principal angles between low-rank approximations of $\mathcal{L}_{M,50}(\mathcal{H}, \mathcal{F})$ and $\mathcal{L}_{M',50}(\mathcal{H}, \mathcal{F})$ for each pair $(M, M')$ of models described in Section B.1.1. For these matrices, we used the same sets $\mathcal{H}, \mathcal{F}$ deriving from the `wiki` dataset as were used in Sections B.1.2 and B.1.3. Note that all pairs of models $(M, M')$ show significant overlap in the column spaces of their logit matrices, with the exception of when either $M$ or $M'$ is the untrained variant of OLMo-1b. This is unsurprising, given that the latter cannot be expected to capture any semantic information in sequences. Interestingly, there is still a bit more overlap than would be expected from random subspaces. We believe this may be because of the nature of the transformer architecture: for instance, if two sequences are very close (e.g., differ in a single token), then it is reasonable to expect that the next-token distributions induced by even a transformer with random weights are somewhat close, due to the nature of attention. We leave further investigation of this point to future work.

## B.3 LINEAR GENERATION

The experiments using LINGEN in Section 3.3 make use of OLMo-1b and generate for a total of 15 tokens. Below we provide details pertaining to how the coefficient vector $v$ was chosen for each of **Option 1** and **Option 2** described in Section 3.3:

**Option 1.** For this case, we chose 40000 histories and 10000 futures from the `wiki` dataset, as described in Section B.1.1. We chose the lengths $\ell_{\min} = 8, \ell_{\max} = 30$, but now measured the lengths with respect to *tokens* (not characters), since we only intend to use the resulting histories and futures for a single model. As in the previous sections, because the token space $\Sigma$ is extremely large, we cannot perform the regression described in Section 3.3 on the entire logit matrix $\mathcal{L}_M(\mathcal{H}, \mathcal{F})$, so we instead used the submatrix $\mathcal{L}_{M,50}(\mathcal{H}, \mathcal{F})$ as described in Section 3.1.

**Option 2.** For this case, we generated sets $\mathcal{H}, \mathcal{F}$ of histories and futures as in **Option 2**, but then randomly permuted all tokens amongst all elements of each of $\mathcal{H}$ and $\mathcal{F}$, which yielded sets $\mathcal{H}^{\text{nonsense}}, \mathcal{F}^{\text{nonsense}}$. The remaining details of the experiment are identical to **Option 1**.

**Baselines.** In Figures 1b and 6 we compare the performance of LINGEN to the following baselines:

- **Single-token version of LINGEN**: this may be viewed as using LINGEN but where the matrix $\mathcal{L}_{M,50}(\mathcal{H}, \mathcal{F})$ was replaced with $\mathcal{L}_M(\mathcal{H}, \{\text{Null}\})$, i.e., the *full* logit matrix where there is a single future, namely the empty future.

- **Generation from a short context**: we generate from OLMo-1b using a context window of 5 tokens (i.e., we mask out all tokens more than 5 units in the past).

- **Intermediate training checkpoint**: we generate from OLMo-1b using the checkpoint "stage1-step1907359-tokens4001B" which corresponds to the end of Stage-1 pretraining.

---

**Algorithm 1** LINGEN Algorithm

---

1: **Input:** Language model $M$, histories $\mathcal{H} \subset \Sigma^\star$, vector $v \in \mathbb{R}^\mathcal{H}$.
2: **Input:** Parameter $m$ (number of new tokens to generate)
3: **function** $\mathsf{LinGen}_M(\mathcal{H}, v, m)$
4:     **for** $t = 1, 2, \ldots, m$ **do**
5:         Compute $\text{logits}_t = v^\top \cdot \mathcal{L}_M(\mathcal{H}, \{z_{1:i-1}\})$.     ▷ ($\mathcal{L}_M(\mathcal{H}, \{z_{1:i-1}\})$ *is the logits matrix where the set of futures is the singleton sequence* $z_{1:i-1}$.)
6:         Sample next token $z_t \sim \text{softmax}(\text{logits}_t)$
7:     **end for**
8:     **Return:** the sequence $(z_1, \ldots, z_m)$.
9: **end function**

---

| LINGEN | LINGEN with single-token logit matrix |
|---|---|
| *Third Sea Lord was given the command upon taking leave from the Admiralty. He hoisted his flag in "Bac*chon" (1874), which was in the Mediterranean under Rear Admiral | *Third Sea Lord was given the command upon taking leave from the Admiralty. He hoisted his flag in "Bac*chusa in 1881 and 1892 compilation also showed that the |
| *in 2007 in Rajasthan, India. Krishna Bhatt has been twice the* recipient of the Award for literary Excellence and has been to Ghana, Switzerland, | *in 2007 in Rajasthan, India. Krishna Bhatt has been twice the* winner of the Golden Mapuhal Trophy at the General Bob badi of |
| *" and Katy Gibson in "Gigi*" had the world's best-known actors and pop singers Peter O'To | *" and Katy Gibson in "Gigi*" and "Diva" were produced, and they were thus of the |
| *2018 including Jago Wali Raat and her first devotional song Tor Dita Lalan Nu*. For the song God Save India, she uses traditional instruments like Bhajan | *2018 including Jago Wali Raat and her first devotional song Tor Dita Lalan Nu*. Rebecca and Mr. Costa confess to each other that they were falling in |
| *KSTQ (93.5 FM) is a radio station licensed to Stuart, Oklahoma, United States. The station* is owned by KFBL Radio and is currently simulcasting KXLP | *KSTQ (93.5 FM) is a radio station licensed to Stuart, Oklahoma, United States. The station* is locally called "The Rock," is currently operated by Axion and E |

Table 2: Sample generations from LINGEN (**Option 1**) and single-token baseline. *Prompts are shown in gray and italicized,* and continuations are shown in black. Samples were not cherry-picked, i.e., the first generation from each prompt was selected. The generations from LINGEN (left) all read well and make clear use of context in the prompt; in contrast, while generations from LINGEN with the single-token logit matrix (right) typically are reasonable for the first token or few, they become derailed soon thereafter.

| LINGEN generations with $\mathcal{H}^{\text{nonsense}}$ |
| --- |
| *Third Sea Lord was given the command upon taking leave from the Admiralty. He hoisted his flag in "Bac*chill" confusing manoeuvre that involved making the windward course. |
| *Third Sea Lord was given the command upon taking leave from the Admiralty. He hoisted his flag in "Bac*ch" for the first time. After Sapp didn't take part in |
| *Third Sea Lord was given the command upon taking leave from the Admiralty. He hoisted his flag in "Bac*chel" and joined with her and the USS Vixen, a |
| *in 2007 in Rajasthan, India. Krishna Bhatt has been twice the* National Head of Dairy Labelling 2004 and 2014. He |
| *in 2007 in Rajasthan, India. Krishna Bhatt has been twice the* national shortlisting for the International Youth Olympic Festival (IYOOF) |
| *in 2007 in Rajasthan, India. Krishna Bhatt has been twice the* author of a book entitled 'Chaerbato Jaataa Aap |
| *" and Katy Gibson in "Gigi*". Consists of: -lrb- lead -rrb- |
| *" and Katy Gibson in "Gigi*", for which she received a nomination at the 2020 Jogja Music |
| *" and Katy Gibson in "Gigi*". Music career. In addition to her television work, Bryant has released several |
| *2018 including Jago Wali Raat and her first devotional song Tor Dita Lalan Nu* Ghareba Hae. A new format was created by exposing a new |
| *2018 including Jago Wali Raat and her first devotional song Tor Dita Lalan Nu*, Ranthi Narayan Thakkar are known for their voice and |
| *2018 including Jago Wali Raat and her first devotional song Tor Dita Lalan Nu* Ram". She made her film debut opposite Satish Kekar in Raj |
| *KSTQ (93.5 FM) is a radio station licensed to Stuart, Oklahoma, United States. The station* is a 24/7 transmitter reports that broadcasts to Tulsa, Page, |
| *KSTQ (93.5 FM) is a radio station licensed to Stuart, Oklahoma, United States. The station* is locally owned by Quad Cities Radio and is maintained by Educational Insights. In |
| *KSTQ (93.5 FM) is a radio station licensed to Stuart, Oklahoma, United States. The station* is a reading of a Victim News radio station licensed to Tulsa, Oklahoma broadcasts |

Table 3: Sample generations from LINGEN and single-token baseline with $\mathcal{H}^{\text{nonsense}}$ (i.e., **Option 2**), for the same 5 target prompts as in Table 2. *Prompts are shown in gray and italicized*, while LINGEN's generations are in black. While not all of the generations are natural continuations of the target prompt, most of the generations show clear use of information from the prompt.

## C  COMPARISON TO NATURAL BASELINES

In this section, we briefly discuss comparison of the low rankness of the extended logit matrix to natural baselines. First, let us consider a random model for comparison. Recall that the extended logit matrix $\mathcal{L}_{\mathcal{D}}(\mathcal{H}, \mathcal{F})$ is a $|\mathcal{H}| \times |\mathcal{F}| \cdot |\Sigma|$ matrix. A matrix of the same dimension with i.i.d. Gaussian entries, denoted by $G$ (i.e., $G_{ij} \sim \mathcal{N}(0, 1)$), has a well-understood asymptotic spectrum given by the Marchenko-Pastur law. Further, non-asymptotic bounds on the singular values of $G$ are also known (See e.g. (Rudelson & Vershynin, 2010)). In particular, the results suggest that the singular values of $G$ are all of the order $\Theta(\sqrt{n})$ where $n = \max\{|\mathcal{H}|, |\mathcal{F}| \cdot |\Sigma|\}$ with even the deviation of the smallest singular value from $\sqrt{n}$ being of the order $\Theta(n^{1/6})$. This indicates that the singular values of $G$ do not decay rapidly and are all of the same order and hence $G$ is not approximated well by a low-rank matrix.

More, interestingly in order to observe power law decays in the singular values, as noted empirically in Section 3.1, one needs to consider matrices with dependent entries. An interesting future direction suggested by our work is to potentially find interesting random matrix models that can be both used to explain the power law decay in the singular values of the extended logit matrix, while also helping us understand the underlying structure in natural language.

A second natural baseline to compare the low-rankness of the extended logit matrix that arise purely from the low-rank structure of the single step logit matrix. To flesh this out consider, the matrix $L = \mathcal{L}_M(\mathcal{H}, \mathcal{F})$ for $\mathcal{H} = \Sigma^t$ and $\mathcal{F} = \text{Null}$ and set $\tilde{L} = \mathcal{L}_M(\mathcal{H}, \mathcal{F})$ for $\mathcal{H} = \Sigma^{t/2}$ and $\mathcal{F} = \Sigma^{t/2}$. The choice of $t/2$ was made for simplicity, and the argument can be easily extended to other splits of $t$. Now, from the argument regarding the low-rankness of the single step logit matrix, we have that $\text{rank}(L) \leq d$ but for the sake of argument let us assume that $d = 1$. This indicates that the rows of $L$ are all multiples of each other, again for simplicity assume that it is a constant multiple of $e_1$ the first standard basis vector. Understanding the struture of $\tilde{L}$ tells us that the rows of $\tilde{L}$ are concatenations of the rows of $L$ corresponding to splits of the history into two halves. But, since the coefficient corresponding to each row of $L$ was arbitary, the rank of $\tilde{L}$ corresponds to the rank of an arbitrary matrix of dimension $|\Sigma|^{t/2} \times |\Sigma|^{t/2}$ which is exponentially large in $t$. Thus, the low-rankness of the single step logit matrix does not imply any non-trivial upper bound on the rank of the extended logit matrix. In particular, this can be seen as a strong argument for the claim that low-rankedness of the extended logit matrix is an extremely surprising phenomena that points towards the presence of interesting structure in natural language and models thereof.

## D  DEFERRED PROOFS OF THEORETICAL RESULTS

As a reference, we first state the definition of an Input Switched Affine Network (ISAN) by Foerster et al. (2017). Note that the difference compared to a time-varying ISAN in Theorem 4.2 is that the transitions depend only on the current token but not on the timestep.

**Definition D.1** (Input Switched Affine Network (ISAN)). *An Input Switched Affine Network (ISAN) is defined by matrices $A_z \in \mathbb{R}^{d \times d}$ for each $z \in \Sigma$ and a matrix $B \in \mathbb{R}^{\Sigma \times d}$ and an initial state $x_0 \in \mathbb{R}^d$. It defines a distribution over sequences of tokens at each timestep $t = 1, 2, \ldots$:*

- *Sampling: token $z_t$ is sampled from $\text{softmax}(Bx_{t-1})$.*

- *State Update: the hidden state is updated as $x_t = A_{z_t} x_{t-1}$.*

First, we show an equivalence between an ISAN and a time-varying ISAN up to a factor of $T$ in the hidden dimension.

**Fact D.2** (Reducing a Time Varying ISAN to an ISAN). *For any time-varying ISAN with hidden dimension $d$ that generates a distribution over sequences of tokens of length $T$, there exists an ISAN with hidden dimension $Td$ that generates the same distribution.*

*Proof.* Let the initial state of the time-varying ISAN be $x_0 \in \mathbb{R}^d$ and its transition and observation matrices be $A_{z,t} \in \mathbb{R}^{d \times d}, B_t \in \mathbb{R}^{\Sigma \times d}$ for $z \in \Sigma, t \in [T]$. To express this as a (time-invariant) ISAN, we build a $Td$-dimensional hidden state consisting of $T$ separate $d$-dimensional blocks. We

can then aggregate the transition and observation matrices across the different timesteps by defining for each token $z \in \Sigma$,

$$
A_z = \begin{bmatrix} 0 & 0 & \cdots & 0 & 0 \\ A_{z,1} & 0 & \cdots & 0 & 0 \\ 0 & A_{z,2} & \cdots & 0 & 0 \\ \vdots & \vdots & \ddots & \vdots & \vdots \\ 0 & 0 & \cdots & A_{z,T-1} & 0 \end{bmatrix}
$$
$$
B = [B_1 \quad B_2 \quad \ldots B_T] \, .
$$

We let the initial state be $(x_0, 0, \ldots, 0)$ i.e. $x_0$ on the first $d$ coordinates and $0$ everywhere else. It is immediate by construction that this ISAN is equivalent to the time-varying ISAN because at each timestep $t$, the hidden state $x_t$ of the original time-varying ISAN is simply stored in the $t + 1$st $d$-dimensional block and is moved to the next block after a transition (the state becomes identically $0$ after the last transition). Thus, this ISAN generates the exact same distribution over sequences of length $T$, as desired. $\qquad \square$

### D.1   EQUIVALENCE BETWEEN TIME-VARYING ISAN AND LOG LOGIT RANK

This subsection is devoted to proving Theorem 4.3. We first show one direction, that low logit rank implies expressibility as a time-varying ISAN.

**Lemma D.3** (Low Logit Rank Implies Time-Varying ISAN). *Let $M$ be a language model such that for each $t \in [T]$, the matrix $\mathcal{L}_M(\Sigma^t, \Sigma^{\leq T-t})$ (recall Theorem 2.2) has rank $d$. Then there is a time-varying ISAN of hidden dimension $d$ that generates the exact same distribution as $M$ on sequences of length $T$.*

*Proof.* For each length $t = 0, 1, \ldots, T-1$, let $\mathcal{S}^{(t)} \subset \Sigma^t$ be a (multi)set of sequences with $|\mathcal{S}^{(t)}| = d$ such that the rows of $\mathcal{L}_M(\mathcal{S}^{(t)}, \Sigma^{\leq T-t})$ span the row space of $\mathcal{L}_M(\Sigma^t, \Sigma^{\leq T-t})$. In other words, this is saying that the rows indexed by $\mathcal{S}^{(t)}$ span all of the rows indexed by sequences of length $t$. Note that such a set exists by the rank assumption — for $t = 0$ there is only one row, and we allow for a multiset so if $|\Sigma^t| \leq d$ then we can arbitrarily choose to duplicate some of the elements so that $|\mathcal{S}^{(t)}| = d$ exactly.

For each timestep $t \in [T]$ and token $z \in \Sigma$ we can construct a matrix $A_{z,t} \in \mathbb{R}^{d \times d}$ such that

$$
\mathcal{L}_M(\mathcal{S}^{(t-1)} \circ z, \Sigma^{\leq T-t}) = A_{z,t} \mathcal{L}_M(\mathcal{S}^{(t)}, \Sigma^{\leq T-t}) \tag{2}
$$

where on the LHS, $\mathcal{S}^{(t-1)} \circ z$ denotes the set obtained by appending the token $z$ to each sequence in $\mathcal{S}^{(t-1)}$. The above is possible because each of these are now length $t$ sequences in $\Sigma^t$ and we assumed that the rows of $\mathcal{L}_M(\mathcal{S}^{(t)}, \Sigma^{\leq T-t})$ span all of these. Also, let $B_t \in \mathbb{R}^{\Sigma \times d}$ be the matrix $B_t = \mathcal{L}_M(\mathcal{S}^{(t)}, \{\text{Null}\})^\top$ where here the set of futures consists of only the empty string.

Now we claim that the ISAN defined by $A_{z,t}, B_t$ for $z \in \Sigma, t \in [T]$ and initial state $x_0 = e_1$ (where $e_1$ is the first standard basis vector) exactly samples from the distribution $M$ over sequences of length $T$. We first show by induction that if $z_{1:t}$ is the sequence sampled so far after $t$ timesteps, then the hidden state satisfies

$$
x_t^\top \mathcal{L}_M(\mathcal{S}^{(t)}, \Sigma^{\leq T-t}) = \mathcal{L}_M(\{z_{1:t}\}, \Sigma^{\leq T-t}) \tag{3}
$$

for $t = 0, 1, \ldots, T$. Note that there is only one empty string so the base case for $t = 0$ is obvious. Now given the inductive hypothesis for $t$, we immediately have that for any choice of token $z_{t+1}$,

$$
x_t^\top \mathcal{L}_M(\mathcal{S}^{(t)} \circ z_{t+1}, \Sigma^{\leq T-t-1}) = \mathcal{L}_M(\{z_{1:t+1}\}, \Sigma^{\leq T-t-1})
$$

just from Theorem 2.2, since the above is just selecting a subset of the columns of the equality in (3). Now since $x_{t+1} = A_{z_{t+1},t+1}^\top x_t$, combining with (2) gives

$$
x_{t+1}^\top \mathcal{L}_M(\mathcal{S}^{(t+1)}, \Sigma^{\leq T-t-1}) = \mathcal{L}_M(\{z_{1:t+1}\}, \Sigma^{\leq T-t-1})
$$

completing the induction. Next, it remains to show that given (3) for all $t$, that each token is sampled from the same conditional distribution as $M$ given the prefix $z_{1:t}$. To see why this is the case, (3) implies that

$$
x_t^\top B_t^\top = x_t^\top \mathcal{L}_M(\mathcal{S}^{(t)}, \{\text{Null}\}) = \mathcal{L}_M(\{z_{1:t}\}, \{\text{Null}\})
$$

since this is just obtained by selecting a subset of the columns in (3). Note that $\mathcal{L}_M(\{z_{1:t}\}, \{\text{Null}\})$ is exactly the mean-centered next token logits conditioned on $z_{1:t}$, and so for both the underlying distribution $M$ and the ISAN we defined, conditioned on the prefix $z_{1:t}$, the next token is sampled according to $\text{softmax}(B_t x_t)$. Thus, the ISAN generates the same distribution as $M$, completing the proof. $\qquad\square$

Now we prove the reverse direction that a time-varying ISAN expresses a distribution with low logit rank.

**Fact D.4** (Time-Varying ISAN has Low Logit Rank). *Let $M$ be a time-varying ISAN with hidden dimension $d$ that generates some distribution over sequences of tokens of length $T$. Then for any $t \in [T]$, the matrix $\mathcal{L}_M(\Sigma^t, \Sigma^{\leq T-t})$ (recall Theorem 2.2) has rank at most $d$.*

*Proof.* For any sequence $h \in \Sigma^t$, let $x_h$ be the hidden state of the ISAN at timestep $t$, after outputting the sequence of tokens $h$. Now consider any future $f \in \Sigma^{\leq T-t}$. Let the tokens of $f$ be $z_{t+1}, \ldots, z_{t+k}$. Then the vector of mean-centered logits for the next token conditioned on $h \circ f$, given by $\{L_M(z|h \circ f)\}_{z \in \Sigma}$ (recall Theorem 2.1), can be obtained by taking the vector

$$BA_{z_{t+k}, t+k} \cdots A_{z_{t+1}, t+1} x_h$$

and then subtracting the mean from all of the entries. In other words, these logits are a linear function of $x_h$ and there is a matrix $A_f \in \mathbb{R}^{\Sigma \times d}$ such that $A_f x_h$ gives exactly the mean-centered logits $\{L_M(z|h \circ f)\}_{z \in \Sigma}$. This implies, by Theorem 2.2, that we can write

$$\mathcal{L}_M(\Sigma^t, \Sigma^{\leq T-t}) = \begin{bmatrix} x_{h_1}^\top \\ x_{h_2}^\top \\ \vdots \end{bmatrix} \begin{bmatrix} A_{f_1}^\top & A_{f_2}^\top & \ldots \end{bmatrix}$$

where the rows of the first matrix are indexed by all possible histories $h \in \Sigma^t$ and the second matrix is a block matrix with blocks indexed by futures $f \in \Sigma^{\leq T-t}$. The above factorization implies that $\mathcal{L}_M(\Sigma^t, \Sigma^{\leq T-t})$ has rank at most $d$, as desired. $\qquad\square$

**Remark 1** (Importance of Mean-centered Logits). *Note that the mean-centering of the logits is important for Fact D.4 to be true. Otherwise, if we used, say, normalized logits, then these logits would no longer be a purely linear function of the hidden state $x_h$ due to subtracting the normalizing constant which is not a linear function of the entries (whereas the mean is).*

*Proof of Theorem 4.3.* Theorem 4.3 now follows immediately from combining Lemma D.3 and Fact D.4. $\qquad\square$

## D.2 TIME-VARYING ISAN REPRESENTATION POWER

In this section, we analyze the representation power of the time-varying ISAN, showing that it can represent a variety of simple architectures and languages. First, we show that it can represent linear state space layers, the core building block in modern state space models (Gu et al., 2021a; Gu & Dao, 2023). We begin by formally defining the class of state space models that we consider.

**Definition D.5** (Selective State Space layer). *A selective state space layer maps a sequence of inputs $u_1, \ldots, u_T \in \mathbb{R}^p$ to a sequence of outputs $y_1, \ldots, y_T \in \mathbb{R}^q$ and has a hidden state with dimension $d$. The initial state is $x_0 \in \mathbb{R}^d$ and the state space layer operates according to the following recurrence:*

$$x_t = A(u_t)x_{t-1} + B(u_t)u_t$$
$$y_t = C(u_t)x_{t-1} + D(u_t)u_t$$

*where the matrices $A(u_t) \in \mathbb{R}^{d \times d}, B(u_t) \in \mathbb{R}^{d \times p}, C(u_t) \in \mathbb{R}^{q \times d}, D(u_t) \in \mathbb{R}^{q \times p}$ may depend arbitrarily on the input $u_t$ at timestep $t$.*

**Remark 2** (On Variants and Stacking of SSM Layers). *This is a general form that encompasses most definitions of a selective state space layer — implementations that are used in practice often enforce more structure on the matrices, especially $A$. Earlier, simpler variants of the SSM used time-invariant layers where $A, B, C, D$ are all fixed, independent of $T$.*

*State space layers are often stacked together — although with the above form, a stack of $\ell$ state space layers can be represented as a single layer with dimension $\ell d$ as long as the transitions matrices $A(u_t), B(u_t), C(u_t), D(u_t)$ depend only on the external input (but not on the recursively obtained intermediate inputs to layers in the middle, see [Hespanha (2018)](#)).*

Now it is not difficult to see that an ISAN can represent a language model consisting of a selective state space layer with softmax readout and no additional nonlinearities (where the tokens are embedded and then fed in as the inputs $u_t$) as defined below.

**Definition D.6** (Linear-in-state SSM)**.** *A linear-in-state SSM defines a distribution over sequences in $\Sigma^T$ as follows. There is a selective state space layer with input, output and hidden state dimensions $p, q, d$. The output tokens are obtained by sampling $z_t \sim softmax(U y_t)$ where $U \in \mathbb{R}^{\Sigma \times q}$ is the readout matrix and the next input is obtained as $u_{t+1} = V e(z_t)$ where $e(z_t) \in \mathbb{R}^{\Sigma}$ denotes the one-hot encoding of $z_t$ and $V \in \mathbb{R}^{p \times \Sigma}$ is the embedding matrix. The initial input $u_1$ is some fixed vector.*

We now prove the following fact:

**Fact D.7** (Representing Linear-in-state SSMs)**.** *A Linear-in-state SSM with input, output, and hidden dimension $p, q, d$ can be expressed as an ISAN with hidden dimension $d + 2q + 1$[8].*

*Proof of [Fact D.7](#).* We construct an ISAN that at each timestep stores $h_{t-1} = (x_t, C(u_t)x_{t-1}, D(u_t)u_t, 1)$ as its hidden state. Note that the dimension of this is $d + 2q + 1$. Now by definition, the next token $z_t$ is sampled from $softmax(U(C(u_t)x_{t-1} + D(u_t)u_t))$ and the vector inside the softmax is clearly a linear function of the hidden state. Next, once we fix $z_t$, the next input $u_{t+1}$ is fixed and we have $x_{t+1} = A(u_{t+1})x_t + B(u_{t+1})u_{t+1}$, and thus the next hidden state $h_t = (x_{t+1}, C(u_{t+1})x_t, D(u_{t+1})u_{t+1}, 1)$ is a linear function of the previous one once $u_{t+1}$ is fixed. This implies that the entire linear-in-state SSM can be expressed as an ISAN, completing the proof. $\square$

Next, we will also show that the ISAN model can express basic languages and algorithmic behaviors. We show that it can solve the task of copying, i.e. generating sequences of the form $a \circ a$ where $a$ is a random bitstring. This task is a standard benchmark for testing the ability of sequence models to capture long-range dependencies and has been used to separate the power of different architectures ([Jelassi et al., 2024](#)).

First, we formalize the task of copying as being able to sample from the following distribution.

**Definition D.8** (Copying)**.** *Define the $n$-bit "copying" distribution as the distribution over length $2n$ sequences of the form $a \circ a$ where $a \in \{0, 1\}^n$ is uniformly random.*

Now we prove that ISANs can express copying.

**Fact D.9** (Representing Copying)**.** *There is a time-varying ISAN with hidden dimension $n + 1$ that generates a distribution arbitrarily close in TV distance to the $n$-bit copying distribution.*

*Proof.* Initialize the hidden state $x_0$ to $n$ zeros and the last bit is $1$. Now for the first $n$ steps, the output matrices $B_1, \ldots, B_n$ are all $0$ (so the output is uniformly random) and the transition matrices simply store the first $n$ bits generated in the first $n$ bits of the state while maintaining that the last bit is $1$. Now for the next $n$ steps, the transition matrices are identity and the output matrices $B_{n+1}, \ldots, B_{2n}$ are

$$B_{n+j} = \begin{bmatrix} 0 & \ldots & 0 & \ldots & 0 & C/2 \\ 0 & \ldots & C & \ldots & 0 & 0 \end{bmatrix}$$

where the last column is always the same and the column with $C$ is the $j$th column and $C$ can be chosen arbitrarily large. Then if the $j$th bit of the state is $0$, then the output will be $0$ with $\exp(C/2)/(1 + \exp(C/2))$ probability and if the $j$th bit of the state is $1$, then the output will be $1$ with $\exp(C/2)/(1 + \exp(C/2))$ probability. Since $C$ can be chosen arbitrarily large, this ISAN can generate a distribution that is arbitrarily close in TV distance to the copying distribution. $\square$

---

[8]This result can also be applied to a stack of such layers, as long as we only allow the transitions to depend on the external input, but not the intermediate values.

Finally, we show that the ISAN model can generate from a noisy parity distribution. Noisy parity is a fundamental problem that has been extensively studied in computational complexity, learning theory, and cryptography. In fact, the hardness of learning a noisy parity from samples is a standard cryptographic assumption (Blum et al., 2003; Pietrzak, 2012). First, we formally define a noisy parity distribution:

**Definition D.10** (Noisy Parity). *A noisy parity distribution over $\{0,1\}^{n+1}$ is defined by a sequence $y \in \{0,1\}^n$ and probability $p \in (0, 1/2)$ and is obtained by drawing $z \in \{0,1\}^n$ uniformly at random and outputting $(z, b)$ with $b \in \{0,1\}$ defined as*

$$b = \begin{cases} \langle y, z \rangle \mod 2 \text{ with probability } 1 - p \\ 1 - \langle y, z \rangle \mod 2 \text{ with probability } p \end{cases}$$

Now we prove that ISANs can express any noisy parity.

**Fact D.11** (Representing Noisy Parity). *For any noisy parity distribution over $\{0,1\}^{n+1}$ , there is a time-varying ISAN with hidden dimension $2$ that exactly generates it.*

*Proof.* We claim that for $M$ being a noisy parity distribution, matrices $\mathcal{L}_M(\{0,1\}^t, \{0,1\}^{\leq n+1-t})$ have rank at most 2 for all $0 \leq t \leq n+1$. To see this, we simply observe that each of these matrices has at most two distinct rows — the only information in the first $t$ bits that matters is the parity of the inner product with $y$ restricted to these first $t$ bits. Now we apply Theorem 4.3 and this completes the proof. $\square$

The fact that ISANs can represent noisy parities, combined with the following standard cryptographic assumption, implies that it is computationally hard to learn an ISAN from samples, which we state in Theorem D.13.

**Assumption D.12.** *[Hardness of Learning a Noisy Parity] There exists no polynomial time algorithm that given access to samples from a noisy parity distribution with hidden vector $y$ can recover a vector $\hat{y}$ that is non-trivially correlated with $y$ (say $|y \oplus \hat{y}| < n/4$).*

**Theorem D.13.** *Under Assumption D.12, there is no polynomial time algorithm that given access to samples from a time-varying ISAN can learn a distribution that is close in total variation distance.*

### D.3    LEARNING AN ISAN FROM LOGIT QUERIES

In this subsection, we present the details of our algorithm for learning a time-varying ISAN from logit queries. First, we formalize the query model.

**Definition D.14** (Logit Query). *The learner can specify any prefix $h \in \Sigma^*$ and obtain the mean-centered logits $L_M[z|h]$ for $z \in \Sigma$.*

Observe that logit queries can simulate samples from the language model. We do this by repeatedly getting the next token logits and then taking softmax to sample the next token and then repeat.

**Fact D.15.** *Given logit query access to $M$, given any history $h \in \Sigma^*$, we can sample a future $f$ of any length from the distribution $\Pr_M[f|h]$.*

**Definition D.16** (Slicing Notation). *For any $t \leq T$, we will use the notation $M[:t]$ to denote the distribution of length $t$ prefixes for sequences drawn from $M$.*

Now we present and analyze the algorithm for learning from logit queries. At a high level, the algorithm tries to construct "spanning" sets of histories and futures $\mathcal{H}_t, \mathcal{F}_t$ for $t = 0, 1, \ldots, T-1$. Ideally, we would be able to ensure that the matrix $\mathcal{L}_M(\mathcal{H}_t, \mathcal{F}_t)$ has rank equal to $\mathcal{L}_M(\Sigma^t, \Sigma^{\leq T-t})$ which would mean that $\mathcal{H}_t, \mathcal{F}_t$ in some sense cover all of the dimensions. Unfortunately, this guarantee is not possible because the rank of $\mathcal{L}_M(\Sigma^t, \Sigma^{\leq T-t})$ could arise from some very rare sequences that will be impossible for us to find algorithmically. We can instead guarantee a weaker notion where we cover all dimensions that are not too rare. The algorithm, described by the subroutine in Algorithm 3, iteratively constructs the sets $\mathcal{H}_t, \mathcal{F}_t$ by adding new sequences that increase the rank of $\mathcal{L}_M(\mathcal{H}_t, \mathcal{F}_t)$ until no such sequences can be found. The key to the analysis is defining the right notion of coverage so as to ensure that the algorithm terminates in polynomial time (see Lemma D.17). We then complete the proof by showing that with this guarantee, we can construct an ISAN that is $\epsilon$ close to the true distribution $M$ in TV distance.

---

**Algorithm 2** Learning from Logit Queries for (time-varying) ISAN

---

1: **Input:** logit query access to time-varying ISAN $M$
2: Initialize $\mathcal{H}_t = \{y^t\}$ for an arbitrary $y \in \Sigma$ for $t = 0, 1, \ldots, T-1$ (where $y^0 = $ Null)
3: Initialize $\mathcal{F}_t = \{$Null$\}$ for $t = 0, 1, \ldots, T-1$
4: Set $N = 4T/\epsilon \log(dT/\epsilon)$
5: Set $\{\mathcal{H}_t\}_t, \{\mathcal{F}_t\}_t \leftarrow \mathsf{CompleteSpan}_M(\{\mathcal{H}_t\}_t, \{\mathcal{F}_t\}_t, N)$
6: Set initial state to be $\widehat{x_0} \in \mathbb{R}^d$ as $x_0 = (1, 0, \ldots, 0)$
7: **for** $t \in [T-1], z \in \Sigma$ **do**
8:     Solve for $\widehat{A_{z,t}} \in \mathbb{R}^{|\mathcal{H}_t| \times |\mathcal{H}_{t-1}|}$ such that

$$\mathcal{L}_M(\mathcal{H}_{t-1} \circ z, \mathcal{F}_t) = \widehat{A_{z,t}}^\top \mathcal{L}_M(\mathcal{H}_t, \mathcal{F}_t).$$

9: **end for**
10: **for** $t \in [T]$ **do**
11:     Set $\widehat{B_t} = \mathcal{L}_M(\mathcal{H}_{t-1}, \{$Null$\})^\top$
12: **end for**
13: Pad matrices $\widehat{A_{z,t}}, \widehat{B_t}$ to appropriate sizes ($d \times d, |\Sigma| \times d$ respectively) by adding rows and columns of zeros
14: **Output:** ISAN with parameters $\widehat{x_0}, \widehat{A_{z,t}}, \widehat{B_t}$

---

**Algorithm 3** Complete Span

---

1: **Input:** logit query access to time-varying ISAN $M$
2: **Input:** Sets $\mathcal{H}_t, \mathcal{F}_t$ for $t = 0, 1, \ldots, T-1$
3: **Input:** Parameter $N$
4: **function** $\mathsf{CompleteSpan}_M(\{\mathcal{H}_t\}_t, \{\mathcal{F}_t\}_t, N)$
5:     Initialize Incomplete $=$ True
6:     **while** Incomplete **do**
7:         Set Incomplete $=$ False
8:         **for** $t \in [T-1]$ **do**
9:             Construct $\mathcal{S}_t$ by taking $N$ i.i.d. samples from $M[:t]$
10:            **if** $\mathrm{Rank}(\mathcal{L}_M(\mathcal{H}_t, \mathcal{F}_t)) < \mathrm{Rank}(\mathcal{L}_M(\mathcal{H}_t \cup \mathcal{H}_{t-1} \circ \Sigma \cup \mathcal{S}_t, \mathcal{F}_t \cup \Sigma \circ \mathcal{F}_{t+1}))$ **then**
11:                Set Incomplete $=$ True
12:                Find elements $h \in \mathcal{H}_{t-1} \circ \Sigma \cup \mathcal{S}_t$ and $f \in \Sigma \circ \mathcal{F}_{t+1}$ such that

$$\mathrm{Rank}(\mathcal{L}_M(\mathcal{H}_t \cup \{h\}, \mathcal{F}_t \cup \{f\})) = \mathrm{Rank}(\mathcal{L}_M(\mathcal{H}_t, \mathcal{F}_t)) + 1$$

13:                Set $\mathcal{H}_t \leftarrow \mathcal{H}_t \cup \{h\}$ and $\mathcal{F}_t \leftarrow \mathcal{F}_t \cup \{f\}$
14:            **end if**
15:         **end for**
16:     **end while**
17:     **Return:** $\{\mathcal{H}_t\}_t, \{\mathcal{F}_t\}_t$
18: **end function**

---

**Lemma D.17.** *Algorithm 3 terminates within $dT$ iterations of the while loop and with $1 - dT^2(1 - \epsilon/(2T))^N$ probability, we have the following property: for any $t \in [T - 1]$, if we draw $h \sim M[: t]$, then with $1 - \epsilon/(2T)$ probability over the randomness of $h$,*

$$Rank(\mathcal{L}_M(\mathcal{H}_t, \mathcal{F}_t)) = Rank(\mathcal{L}_M(\mathcal{H}_t \cup \mathcal{H}_{t-1} \circ \Sigma \cup \{h\}, \mathcal{F}_t \cup \Sigma \circ \mathcal{F}_{t+1})) \quad (4)$$

*where $\mathcal{H}_{t-1} \circ \Sigma$ denotes the set of all possible sequences obtained by concatenating some element of $\mathcal{H}_{t-1}$ with some token in $\Sigma$.*

*Proof.* Note that by induction, we always have $|\mathcal{H}_t| - Rank(\mathcal{L}_M(\mathcal{H}_t, \mathcal{F}_t)) \in \{0, 1\}$ and $|\mathcal{F}_t| - Rank(\mathcal{L}_M(\mathcal{H}_t, \mathcal{F}_t)) \in \{0, 1\}$ (the off-by-one is just because the initial matrix could have rank 0). Note that since $M$ is generated by a time-varying ISAN, by Fact D.4, the rank is always at most $d$. Also in each iteration of the while loop before termination, the sizes of $\mathcal{H}_t, \mathcal{F}_t$ increase by 1 for at least one $t$. Thus, the algorithm can execute at most $dT$ iterations of the while loop before termination.

Next, to prove the second part of the lemma, whenever the desired property doesn't hold for some $t \in [T - 1]$, then with probability at least $1 - (1 - \epsilon/(2T))^N$, the check in the while loop will fail and the flag Incomplete will be set to True. Thus, union bounding over every time we check the rank condition, the overall failure probability is at most $dT^2(1 - \epsilon/(2T))^N$. $\square$

Now we can complete the proof of Theorem 4.4.

*Proof of Theorem 4.4.* First, note that the ISAN we construct in Algorithm 2 is well defined because whenever Algorithm 3 terminates, we must have

$$Rank(\mathcal{L}_M(\mathcal{H}_t, \mathcal{F}_t)) = Rank(\mathcal{L}_M(\mathcal{H}_t \cup \mathcal{H}_{t-1} \circ \Sigma, \mathcal{F}_t \cup \Sigma \circ \mathcal{F}_{t+1}))$$

and thus the matrix $\widehat{A_{z,t}}$ must exist because the rows of $\mathcal{L}_M(\mathcal{H}_t, \mathcal{F}_t)$ must span the rows of $\mathcal{L}_M(\mathcal{H}_{t-1} \circ z, \mathcal{F}_t)$. Also note that all of the operations in Algorithm 2 can be implemented in $\text{poly}(d, |\Sigma|, T, 1/\epsilon)$ time and queries.

Now assuming that the property in Lemma D.17 holds for the output of the CompleteSpan step, we claim that we have $D_{\text{TV}}(M, M') \leq 0.5\epsilon$. For each $t$, let the good set $\mathcal{G}_t \subset \Sigma^t$ be the subset of histories $h$ for which (4) holds.

Now sample token by token according to the ISAN we learn with parameters $\widehat{x_0}, \widehat{A_{z,t}}, \widehat{B_t}$. At timestep $t = 0$, by construction, we have

$$\widehat{x_0}^\top \mathcal{L}_M(\mathcal{H}_0, \mathcal{F}_0) = \mathcal{L}_M(\{\text{Null}\}, \mathcal{F}_0)$$

and our current string is Null. Now assume that at timestep $t$, we have sampled the $t$ tokens $z_{1:t}$ and the current hidden state $\widehat{x_t}$ satisfies

$$\widehat{x_t}^\top \mathcal{L}_M(\mathcal{H}_t, \mathcal{F}_t) = \mathcal{L}_M(z_{1:t}, \mathcal{F}_t).$$

Then since Null $\in \mathcal{F}_t$, we get that softmax$(\widehat{B_t}\widehat{x_t})$ exactly samples the next token with the correct probabilities. Let the next token be $z_{t+1}$. If $z_{1:t} \in \mathcal{G}_t$, then the above equality also implies

$$\widehat{x_t}^\top \mathcal{L}_M(\mathcal{H}_t, z_{t+1} \circ \mathcal{F}_{t+1}) = \mathcal{L}_M(z_{1:t}, z_{t+1} \circ \mathcal{F}_{t+1})$$

which is the same as saying

$$\begin{aligned} \mathcal{L}_M(z_{1:t+1}, \mathcal{F}_{t+1}) &= \widehat{x_t}^\top \mathcal{L}_M(\mathcal{H}_t \circ z_{t+1}, \mathcal{F}_{t+1}) \\ &= \widehat{x_t}^\top A_{z_{t+1}, t+1}^\top \mathcal{L}_M(\mathcal{H}_{t+1}, \mathcal{F}_{t+1}) \\ &= \widehat{x_{t+1}}^\top \mathcal{L}_M(\mathcal{H}_{t+1}, \mathcal{F}_{t+1}). \end{aligned}$$

In other words, by induction, as long as all subsequences remain in the good sets $\mathcal{G}_1, \ldots, \mathcal{G}_T$, then every token is sampled according to the correct conditional distribution, matching $M$. By Lemma D.17 and the definition of the good sets, the probability we ever sample outside of a good set is at most $0.5\epsilon$. The way we set $N$ ensures the conclusion of Lemma D.17 holds with probability at least $1 - 0.5\epsilon$ and thus the overall expected TV distance between $M'$ and $M$ is at most $\epsilon$, as desired. $\square$

## D.4 GENERALIZATION BOUNDS

In this section, we show a generalization bound for when the LinGen algorithm can guarantee high accuracy. We assume that we have fixed the language model $M$, we have a target history $h_0$, and we will approximate $h_0$ as a linear combination of "basis" histories $\mathcal{H} = \{h_1, \ldots, h_n\}$. We then generate $k$ tokens according to LinGen (Algorithm 1).

Lemma D.18 allows us to relate the linear regression error when we draw futures from an arbitrary distribution to the KL divergence between the LinGen distribution and the ground truth distribution for continuations of $h_0$, as long as we have the matrix ordering inequality in (5). The main point is that (5) and (6) (the regression error) can be tested empirically from samples because of matrix concentration (see Fact D.20). Lemma D.18 then shows that if these quantities are sufficiently small, then we can bound the KL divergence between the true distribution and the one obtained by generating according to LinGen.

While our theoretical bounds are not tight enough to directly apply, we still believe this is a worthwhile conceptual point that generalization can be bounded in terms of some testable quantitative "subspace overlap" that is related to the subspace overlap for different distributions of futures $\mathcal{F}, \mathcal{F}'$ discussed in Section 3.2. We also discuss below how we expect tighter bounds to hold in practice due to better concentration than the "worst-case" theoretical analysis.

**Lemma D.18.** *Define the distributions $\mathcal{P}_1$ and $\mathcal{P}_2$ as follows:*

- *$\mathcal{P}_1$ is obtained by uniformly drawing $t \in \{0, 1, \ldots, k-1\}$ and then sampling uniformly at random from $\Sigma^t$*

- *$\mathcal{P}_2$ is obtained by uniformly drawing $t \in \{0, 1, \ldots, k-1\}$ and then sampling a continuation to $h_0$ of length $t$, according to $M$*

*Let $\mathcal{H}_0 = \mathcal{H} \cup \{h_0\}$. Assume that for some parameters $\alpha, \gamma$,*

$$
\begin{aligned}
&\mathbb{E}_{f \sim \mathcal{P}_2}\left[\mathcal{L}_M(\mathcal{H}_0, \{f\})\mathcal{L}_M(\mathcal{H}_0, \{f\})^\top\right] \\
&\preceq \alpha \mathbb{E}_{f \sim \mathcal{P}_1}\left[\mathcal{L}_M(\mathcal{H}_0, \{f\})\mathcal{L}_M(\mathcal{H}_0, \{f\})^\top\right] + \gamma I_{n+1}.
\end{aligned}
\tag{5}
$$

*Then for any vector $v$ such that*

$$
\mathbb{E}_{f \sim \mathcal{P}_1}\left[\left\|\mathcal{L}_M(\{h_0\}, \{f\}) - v^\top \mathcal{L}_M(\mathcal{H}, \{f\})\right\|^2\right] \leq \Delta,
\tag{6}
$$

*if we define*

- *$\mathcal{P}_{\mathsf{LinGen}}$ is the distribution obtained by sampling from $\mathsf{LinGen}(v, \mathcal{H}, k)$*

- *$\mathcal{P}_{\mathsf{truth}}$ is the distribution of length $k$ continuations of $h_0$ according to $M$*

*then*

$$
D_{\mathrm{KL}}(\mathcal{P}_{\mathsf{truth}} \| \mathcal{P}_{\mathsf{LinGen}}) \leq 2k\sqrt{\alpha\Delta + \gamma(1 + \|v\|^2)}.
$$

*Proof.* Let $u$ be the $n+1$-dimensional vector $u := (1, v)$. Taking (5) and taking the inner product of both sides with $uu^\top$ and then applying (6), we have

$$
\mathbb{E}_{f \sim \mathcal{P}_2}\left[\left\|\mathcal{L}_M(\{h_0\}, \{f\}) - v^\top \mathcal{L}_M(\mathcal{H}, \{f\})\right\|^2\right] \leq \alpha\Delta + \gamma(1 + \|v\|^2).
\tag{7}
$$

Now assume that LinGen has sampled a sequence $f = z_1 z_2 \ldots z_t$ so far. Then by the definition of LinGen we can bound

$$
\begin{aligned}
D_{\mathrm{KL}}\left(\Pr_M[\cdot|h_0 \circ f] \,\middle\|\, \Pr_{\mathsf{LinGen}(v, \mathcal{H}, k)}[\cdot|f]\right) &\leq \max_z \log \Pr_M[z|h_0 \circ f] - \log \Pr_{\mathsf{LinGen}(v, \mathcal{H}, k)}[z|f] \\
&\leq 2\max_z \left|L_M[z|h_0 \circ f] - L_{\mathsf{LinGen}(v, \mathcal{H}, k)}[z|f]\right| \\
&\leq 2\left\|\mathcal{L}_M(\{h_0\}, \{f\}) - v^\top \mathcal{L}_M(\mathcal{H}, \{f\})\right\|
\end{aligned}
\tag{8}
$$

where the arguments on the LHS are distributions over the token space $\Sigma$ given by the next-token probabilities in the two different sampling procedures. Now using the KL chain rule, we get

$$D_{\mathrm{KL}}(\mathcal{P}_{\text{truth}}\|\mathcal{P}_{\text{LinGen}}) = k\mathbb{E}_{f\sim\mathcal{P}_2}\left[D_{\mathrm{KL}}\left(\mathrm{Pr}_M[\cdot|h_0\circ f]\,\middle\|\,\mathrm{Pr}_{\text{LinGen}(v,\mathcal{H},k)}[\cdot|f]\right)\right]$$

$$\leq 2k\sqrt{\alpha\Delta + \gamma(1 + \|v\|^2)}$$

where the last step follows by combining (7), (8) and Cauchy-Schwarz. This completes the proof. $\square$

**Remark 3** (Improving the Bound by a Factor of $|\Sigma|$)**.** *The above proof can be "lossy", costing an extra factor of $\sqrt{|\Sigma|}$ when translating from KL to $L^2$ distance in (8). This factor can be saved, provided that the errors in the logits are roughly uniformly distributed over the token space — this is what we observe in practice.*

We can also flip the arguments of the KL divergence to match the quantity we measured empirically in Section 3.3, provided that the logits of the true language model are bounded.

**Corollary D.19.** *Assume that $|L_M[z|h]| \leq C$ for any history $h \in \Sigma^*$ and token $z \in \Sigma$. Then under the same conditions as Lemma D.18, we also have*

$$D_{\mathrm{KL}}(\mathcal{P}_{\text{LinGen}}\|\mathcal{P}_{\text{truth}}) \leq 2(1 + k(\log|\Sigma| + 2C))\sqrt{k}\left(\alpha\Delta + \gamma(1 + \|v\|^2)\right)^{0.25}.$$

*Proof.* Note that the assumption implies

$$-\log\mathrm{Pr}_M[z|h] \leq \log|\Sigma| + 2C$$

for any $h \in \Sigma^*, z \in \Sigma$. Now we can simply combine Fact D.21 and Lemma D.18 to get

$$D_{\mathrm{KL}}(\mathcal{P}_{\text{LinGen}}\|\mathcal{P}_{\text{truth}}) \leq (1 + k(\log|\Sigma| + 2C))\sqrt{2D_{\mathrm{KL}}(\mathcal{P}_{\text{truth}}\|\mathcal{P}_{\text{LinGen}})}$$

$$\leq 2(1 + k(\log|\Sigma| + 2C))\sqrt{k}\left(\alpha\Delta + \gamma(1 + \|v\|^2)\right)^{0.25}$$

as desired. $\square$

Finally, we also state concentration bounds for being able to test (5) and (6) from samples.

**Fact D.20.** *Let $\mathcal{H} \subset \Sigma^*$ with $|\mathcal{H}| = n$ and let $\mathcal{P}$ be any distribution over sequences in $\Sigma^*$. Assume that the language model $M$ satisfies the bound $|L_M[\cdot|h]| \leq C$ for any history $h \in \Sigma^*$. Then with $1 - \delta$ probability, if we let $\mathcal{F}$ be a subset of $S \geq |\Sigma|^2 C^4 n^2 (1/\epsilon)^2 \log(1/\delta)$ samples drawn from $\mathcal{P}$, then*

$$\left\|\frac{1}{|\mathcal{F}|}\mathcal{L}_M(\mathcal{H},\mathcal{F})\mathcal{L}_M(\mathcal{H},\mathcal{F})^\top - \mathbb{E}_{f\sim\mathcal{P}}\left[\mathcal{L}_M(\mathcal{H},\{f\})\mathcal{L}_M(\mathcal{H},\{f\})^\top\right]\right\| \leq \epsilon$$

*and consequently, for any $v \in \mathbb{R}^n$,*

$$\left|\frac{1}{|\mathcal{F}|}\left\|v^\top\mathcal{L}_M(\mathcal{H},\mathcal{F})\right\|^2 - \mathbb{E}_{f\sim\mathcal{P}}\left[\left\|v^\top\mathcal{L}_M(\mathcal{H},\{f\})\right\|^2\right]\right| \leq \epsilon\|v\|^2.$$

*Proof.* For the first inequality, note that

$$\mathcal{L}_M(\mathcal{H},\mathcal{F})\mathcal{L}_M(\mathcal{H},\mathcal{F})^\top = \sum_{f\in\mathcal{F}}\mathcal{L}_M(\mathcal{H},\{f\})\mathcal{L}_M(\mathcal{H},\{f\})^\top.$$

Also note $\|\mathcal{L}_M(\mathcal{H},\{f\})\| \leq C\sqrt{n|\Sigma|}$ by assumption, so Matrix Bernstein immediately gives the first inequality. The second inequality follows directly from the first by taking the inner product of the LHS with $vv^\top$. $\square$

**Remark 4.** *As before, we expect much faster concentration in practice e.g. we used a worst case bound on the individual summands when we applied matrix Bernstein, but a much tighter bound, that saves the dependence on $|\Sigma|$, seems to hold empirically because the columns of $\mathcal{L}_M(\mathcal{H},\{f\})$ are generally not too correlated with each other.*

**Fact D.21.** *Let $P, Q$ be two discrete probability distributions on $s$ elements given by $P = \{p_1, \ldots, p_s\}$ and $Q = \{q_1, \ldots, q_s\}$. Assume that $-\log p_i \geq C$ for all $i \in [s]$. Then*

$$D_{\mathrm{KL}}(Q\|P) \leq (1 + C)\sqrt{2D_{\mathrm{KL}}(P\|Q)}.$$

*Proof.* We can write

$$
\begin{aligned}
D_{\mathrm{KL}}(Q\|P) &= \sum_{i \in [s]} q_i(\log q_i - \log p_i) \\
&\leq \sum_{i \in [s], q_i > p_i} q_i(\log q_i - \log p_i) \\
&\leq \sum_{i \in [s], q_i > p_i} (q_i - p_i)(\log q_i + 1 - \log p_i) \\
&\leq (1 + C) \sum_{i \in [s]} |q_i - p_i| \\
&\leq (1 + C)\sqrt{2D_{\mathrm{KL}}(P\|Q)}.
\end{aligned}
$$

Note that for the second inequality we used the convexity of $x \log x$ and the final step follows from Pinsker's inequality. □

## E    FUTURE DIRECTIONS

Our work motivates numerous directions for future work:

- Can we better understand how the singular value decay evolves during training (recall Figure 3) and can we use it as a diagnostic for training progress?

- The low rank structure allows us to represent each history as a vector (recall Section 1.1) — can we extract concepts and features in this representation space? This would help build towards a model-agnostic approach to interpretability.

- Can we use LINGEN or some variation of it to bypass safety guardrails and generate responses to unsafe prompts? While, as stated, LINGEN requires querying the model on $h_{\mathsf{targ}} \circ f$ for many different choices of $f$, there are avenues for circumventing the need to reveal $h_{\mathsf{targ}}$. For example, we can split up $h_{\mathsf{targ}} = h_1 \circ h_2$, treat $h_1$ as the target history, and then $h_2$ as the first part of the generated future.

- Related to the previous question, does our framework suggest techniques for safeguarding against such attacks?

- Can we extend our theoretical results to the case when the model is only approximately low-rank as observed in practice? Further, it is an interesting open question to understand what notions of approximation (e.g. total variation vs matrix norm) are theoretically feasible while maintaining fidelity to practice.

## F    STATEMENT ON USE OF LLMS

We used LLMs to help us write and refine certain fragments of code e.g. we would provide a description of a function we wanted and ask an LLM to help us implement it. We also used LLMs to help us find related work and references and to help polish the writing after we had a draft.

