# OpenReview forum: "Sequences of Logits Reveal the Low Rank Structure of Language Models"
_ICLR.cc/2026/Conference — ICLR 2026 Oral_

### Official Review · Reviewer_L59i · 2025-10-30

**Soundness:** 4
**Presentation:** 4
**Contribution:** 3
**Rating:** 8
**Confidence:** 2

**Summary:**

This paper proposes a novel, model-agnostic framework for understanding the internal structure of language models by studying their "extended logit matrices." The authors define this matrix based on the log probabilities of sequences (futures) given various contexts (histories).

The paper's contributions are twofold:

- Empirical: They demonstrate across a range of modern LLMs that these extended logit matrices have a consistent, approximate low-rank structure. This structure is shown to emerge early in the pre-training process. This low-rank property is then exploited to create a generation procedure "LINGEN"， which can generate a response to a target prompt by querying the model on unrelated or even nonsensical prompts.

- Theoretical: They connect this empirical finding to a formal generative model, the Input Switched Affine Network (ISAN). They prove that having an exact low logit rank is equivalent to being expressible as a time-varying ISAN and provide provable learning guarantees for this model class under logit query access.

**Strengths:**

- **Novel Framework**: The concept of the "extended logit matrix" is a simple and powerful model-agnostic abstraction. It provides a new lens to study the intrinsic dimensionality and structure of LLMs beyond just analyzing weights or single-token logits.

- **Comprehensive Validation**: The paper does an excellent job supporting its claims with both strong empirical evidence and solid theoretical grounding. The empirical study is thorough, covering multiple model architectures and sizes, and the theoretical connection to ISANs is elegant.

- **Novel Generation & Safety Implications**: The "LINGEN" generation method is a surprising consequence of the observed low-rank structure. The ability to generate text from a target prompt using only queries to unrelated, nonsensical prompts is a significant finding. This has clear and important implications for AI safety, as it suggests a potential new vector for bypassing prompt-based safety filters.

**Weaknesses:**

- **Safety Implications Discussed but Not Fully Explored**: The primary weakness is that the safety implications, while highlighted, are not demonstrated in practice. The paper suggests that LINGEN could be used to "circumvent input filters," but it does not provide a concrete experiment showing such a jailbreak. The analysis is currently a proof-of-concept (i.e., generating coherent text) rather than a practical demonstration of a safety bypass. This is acknowledged as future work (Appendix E), but it leaves the practical severity of this potential vulnerability unclear.

**Questions:**

- Following up on the weakness: Could you elaborate on the practical feasibility of using LINGEN to bypass existing safety alignments? How many histories would be required? Is it feasible to perform the attack on large LLMs? How much cost would it take to construct these histories and their corresponding weights for linear combination?

---

> ### Author Response · Authors · 2025-11-20
> **Response to Reviewer L59i**
>
> Thank you for your helpful review. The main point of this paper is to illustrate a new universal observation about linear structure in LLMs. We view this paper as laying the foundation for understanding LLMs through this low-rank lens. We discuss the safety application as one potential avenue for further exploiting these observations, but the main point of the demonstrations in the paper is to develop a general method and show that it works in principle.
>
>
>
> We agree that a more comprehensive evaluation for actually applying this method to attempt to circumvent input filters would be an interesting direction for future work.  In terms of the number of histories required, currently we are using a basis of  around $10^4$ histories for 1B models.

---

> > ### Comment · Reviewer_L59i · 2025-11-26
> >
> > Thanks for your response, I'll maintain my positive score.

---

### Official Review · Reviewer_PfEp · 2025-10-31

**Soundness:** 4
**Presentation:** 3
**Contribution:** 4
**Rating:** 8
**Confidence:** 4

**Summary:**

The authors present an analysis of low-rank structures for logit matrices in language models. The paper idea is very interesting and new, since previous analyses limit to analyze logit structure for a prompt, here the authors consider logit matrices over sets of histories and futures. The experimental analysis reveals interesting aspects of low-eank structures, which are of extreme interest for the community working in interpreting language models internals.

**Strengths:**

I found that the experimental section is almost excellent and appreciate how the authors give a comprehension of the phenomena observed, state new hypothesis and propose future directions. On its own, this is a great contribution which opens the doors for more future research.

About experiments:
1.  While measuring the complete logit matrix is unfeasible for all tokens and only top-100 are considered, the analysis clearly shows low rank structures across different datasets with histories and futures size $n \in (10^3,10^4)$, both for a logit matrix  $n \times n \times k$ for full set of histories and futures, and also for downsized $n/t \times n/t \times k$, varying $t$ (correct me if I got it wrong).  The scaling of eigenvalues is an interesting observation, which is well explained by the authors through the analysis of  the coefficient $\alpha$.
2. The experimental outcome of principal angles between logit matrix of sensical  (A) and non-sensical futures (A') is also an interesting observation. As explained by the authors, this shows that despite low-rank approximation of A and A',  the two spaces have high intersection, with not much beyond the intersection. The finding that this also holds between different language models is even more interesting, it should be highlighted more in the paper.
3. Using these linear dependencies between histories to control generation with LINGEN is an interesting application and a nice methodological contribution. The only limitation is that LINGEN requires having access to (many) logits of the target history for many futures to evaluate the corresponding $v_{h_{tar}}$. Still, the method is interesting and the authors can highlight more how to improve such limitation in evaluating $v_{h_{tar}}$ (even in a limitation section). In my opinion, a careful choice of futures could help in that direction.

About theory:
* The analysis on time-varying ISAN is valuable, with novel theoretical insights, although with very limited explanation in the text. I am particularly interested in how such a linear model can capture generation in a generic language model, see my questions below.

**Weaknesses:**

While the material is excellent, with new interesting results and a theoretical reformulation of the ISAN model, the clarity should be improved.
My comment arise from the fact that I had to re-read the text many times and continue jumping to appendix before completely capturing the message or familiarizing with the quantities the authors introduce. In fact, while theoretical explanations are useful in the experimental sections, it overloads the reader trying to grasp both the theory and interpret the experiments at the same time.
I suggest the authors to move, as much as possible, theoretical matherial in an apposite section while leaving more space to experimental details alone.

Other points that are weak:
1. **Notation** Denoting with $\mathcal H$ and $\Sigma$ the sets of histories and of tokens, it is a bit imprecise to write $\mathbb R^{\mathcal H}$ and
$\mathbb R^{\Sigma}$ the vector spaces with dimension of the sets. It would be more convenient to use the cardinality or introduce naturals $M$ and $N$ for that. To be clear, this was a bit confusing when reading the manuscript.
2. **Notation** From other papers, feature mappings $\phi, \psi$ are also called the embeddings and unembeddings of the model [1,2].
I also noticed that while $\phi: \mathcal H \to \mathbb R^d$, with $d$ dimension of vector embeddings, $\psi: \Sigma \to \mathbb R^d$.
So it is a bit imprecise to write $\psi(f)$, if $f \in \mathcal F$  is a sentence. Can the authors explain the relation to formalism in [1,2]? \
I noticed that the authors make use of $\phi(h \circ f)$ in Def. 2.2, which makes sense, but suggest that $\psi(f)$ is never used in any step of the computations.
3. **Experiments** Some quantitative results should be made more clear. In 3.2, what is the proportion of "small" principal angles of the two ranks approximations? And how this impacts to finding transfer across different futures? The claim "Thus, linear relationships between histories transfer to significantly different sets of futures." should be substantiated more. In light of this precisation, it would be beneficial to highlight more the results on different language models, which essentially test a different hypothesis (which tells that the vector $v$ of the null space is almost shared between models, if I got it correctly). This aspect should be explained further.
4. **Notation** In lines 359/360 I guess there is a typo in saying $t\geq 1$, since you would have $z_{1:0}$ for $t=1$.
Also denoting with $v$ the vector associated to $h_{tar}$ is a bit confusing. I suggest to denote it as $v_{tar} \in \mathbb R^{|\mathcal H|}.$
5. **Method** LINGEN is interesting, but as far as I can see, estimatingthe vector $v$ requires prompting many times the language model with the history $h_{tar}$. Given the scope of the paper, this is not a serious limitation, but the discussion should be enlarged on this. I expect that some tricks can be created to estimate the vector $v$ for $h_{tar}$ without prompting all the futures. What is the authors' opinion on this matter?
6. The connection to ISAN is interesting but received far too limited space. I could not toatally grasp the message in that section, especially for why a time-varying ISAN would be more preferable than a language model with low-rank. My doubt arises as you need to instantiate $A_{z, t}$ and $B_t$ for each element $t$ in the chain, and each token $z$. Withouth sharing $A$ and $B$ across elements $t$ of the chain, what is the advantage of this construction?  Can you further elaborate? \
There is also a lot of material on ISAN which would constitute a separate contribution and that I could not review from the Appendix. Because of the lenght of the material in the Appendix and the statements not appearing in the main text, I could not verify the correctedness of the claims. I really suggest the authors to take more space to explain the construction and relevant results of ISAN.

**Clarity comment** Overall, some results could be moved to Appendix or shortned. E.g., the whole introduction of KL in Def. 3.1 and discussion can be easily moved to the Appendix without compromising much the text, as they are quite standard,  while leaving space for expanding on some points I mentioned above.

**Comment** Improving clarity would much increase the impact of the authors' contribution, which is extremely timely and relevant for research in language models structure learning. Depending on the rebuttal, I am available to further review a restructuring of the paper with my and other reviewers comments.

-------

[1] Park et al. "The linear representation hypothesis and the geometry of large language models" ICML 2024 \
[2] Marconato et al.  "All or none: Identifiable linear properties of next-token predictors in language modeling" AISTATS 2025

**Questions:**

**Note that:** There is much intersection with works that have treated embeddings and unembeddings in language models. From [1,2], the logit matrix which is used in this paper has a peculiar relation to embeddings and unembeddings. By defining the mean-unembedding vector $\psi_m := \sum_{y \in \Sigma} \psi(y) / |\Sigma|$, then (if I'm not wrong) the logit matrix can be written as:
$$
{L_M}(\mathcal H, \mathcal F)_{(h, (f,z))} = \phi(h \circ f)^\top (\psi(z) - \psi_m)
$$
which is again equal to considering the pivoting step in [2] and similar to as it is done in [3]. Interestingly, this casts a connection to theoretical results in linear properties of language models [1,2] for next-token predictors. Low-rank is new in this sense, but there can be some interesting connection to what are parallel vectors and parallel histories (as you notice in your paper), but also to the effective complexity of the language model [2].

**Q** For the versions of OLMO, what is the representation dimensionality $d$? Did you check if using $d$ components you recover the distribution of original LLM? \
I think that drawing a connection to effective complexity/dimensionality of representations can highlight distinction from the setup in [1,2] which is not concerned with embeddings of futures. This may inspire an update of effective complexity of the model based on histories and futures.

**Q** About results in Sec 3.2, what is the difference in evaluating the principal angles between rank reduced A and A' w.r.t. ones? For rank reduction of A and A' with rank k, if you keep the k-th higher components, should that be the same if you consider rank 2k? I don't understand how this measure depends on rank reduction. Can you further explain?

**Q** What is the authors opinion about the relation to linear properties that are observed for single-token logits? For example, the ISAN construction much resembles relational linearity in a sense. From [2,4,5], the mapping from context to tokens can be seen as relational linear if you have that $z = A_f \phi(h)$, for some specific matrix $A_f$ that only depends on the future $f$. It seems the ISAN model implies something similar for all members of the chain.

------

[3] Hinton et al. "Distilling the knowledge in a neural network" (2015) \
[4] Hernandez et al. "Linearity of relation decoding in transformer language models" ICLR 2024 \
[5] Paccanaro and Hinton "Learning Distributed Representations of Concepts using Linear Relational Embedding" 2020

---

> ### Author Response · Authors · 2025-11-20
> **Response to Reviewer PfEp**
>
> Thank you for your helpful and detailed comments.
>
>
> Point 2 (notation) -- on $\psi(f)$. Indeed, the probabilistic model we are proposing in Section 1.1 (where we write $\psi(f)$) is distinct from the model proposed in [1,2], where the unembeddings $\psi$ take as input just a single token (e.g., as in Eq (1) of [2]). In particular, for our setting, we want the embedding vector $\phi(h)$ to contain all information necessary for generating the entire future $f$, not just the next token. While we do not directly work with this probabilistic model, we use it as inspiration for Definition 2.2: the connection results from the observation that for $f = (f_1, \ldots, f_t)$, we have $\log Pr_M[f | h] = \log Pr_M[f_1 | h] + \cdots + \log Pr[f_t | h \circ f_{1:t-1}]$, and so to understand $\log Pr_M[f | h]$ it suffices to understand $\log Pr_M[y | h \circ f']$ for various sequences $f'$ and tokens $y$.
>
>
>
> Point 3 (experiments). Let us first clarify how we compute the principal angles in Section 3.2. Given two extended logit matrices for two different sets of futures, say $L_{M}(H, F)$ and $ L_{M}(H, F')$ and a parameter $r$ denoting the rank (i.e., corresponding to the $x$-axis of Figure 4), recall that we denote by $A,A'$ the best rank-$r$ approximations to these two logit matrices in Frobenius norm, respectively. Next, we define matrices $U,U'$ as follows: they each have $r$ columns, which consist of orthonormal bases of the column spans of $A,A'$, respectively (we note that $U,U'$ may be computed efficiently by taking an SVD of the logit matrices).
> Finally, we compute the singular values of $U^\top U'$ ---- these are exactly the cosines of the principal angles that we plot.
>
> The way to interpret these values is that if there are $s \leq r$ singular values close to $1$, then there is an $s$-dimensional subspace that is essentially contained in the column spaces of both $A$ and $A'$.  Thus, since the lines in Figure 4 stay close to $1$, only decaying towards $0$ at $s \approx r$, this means that the  column spaces of each of $A$, $A'$ have large overlap.
> Since the column space of a matrix is the orthogonal complement of its row kernel, this overlap means that vectors $v$ that are approximately in the row kernel of $L_M(H,F)$ should also approximately be in the row kernel of $ L_M(H,F')$. This is what we meant by ``transfer across sets of futures.''
>
> Our experiments on LINGEN (with Option 2) indeed give some downstream empirical evidence that linear relationships between histories transfer to significantly different sets of futures: in particular, the coefficient vector $v$ was computed using nonsense sets of futures. Despite this, the future that we generate in response to the target prompt is not nonsense, i.e., it is reasonably well-written English. In other words, using coefficient vectors computed with respect to nonsense futures still yields good results for a prompt whose ``response distribution'' consists of well-formed English.
>
> Finally, regarding the results on different language models, which do test a different hypothesis: we will indeed emphasize this further.
>
> Point 4 (notation on lines 359-360). The sequence $z_{1:0}$ should be interpreted as the empty sequence, i.e., consisting of no tokens.
>
> Point 5 (method). In terms of the limitations of the method, indeed we need to prompt the language model many times on $h_{tar}$. However, one possible workaround is to split up the target prompt into two parts, say $h_{tar} = h_1 \circ h_2$. We then learn the vector $v$ for $h_1$ and then append $h_2$ to all of the basis histories and then run LINGEN starting from there. This would allow the learner to not reveal the entire target prompt to the LLM. We can add a discussion on this  to the paper.
>
> Point 6 (connection to ISAN). The main purpose of introducing the ISAN is that it is a clean parametric generative model that captures the notion of low logit rank. In other words, assuming a language model is low rank does not immediately tell one how to generate from the language model (whereas given an ISAN, it is clear how to generate from it). Our proof of the equivalence tells us how to extract a generative (ISAN) model from any low-logit rank distribution.
>
> In the past, the ISAN was useful in practice as an interpretable recurrent architecture [Foerster et. al. 2017]. In contrast to this prior work, we do not intend for the ISAN definition to be taken literally and trained in practice, but rather we believe it is useful as a simple and mathematically tractable approximation that can provide useful theoretical frameworks for thinking about language models.
>
> Due to the space constraints, we moved the proofs of the theoretical results to the appendix. We mostly intended for the section in the main body to convey that the ISAN is simple and mathematically tractable model that can be a useful proxy for studying theoretical aspects of LLMs.

---

> > ### Author Response · Authors · 2025-11-20
> > **Continuation of Response to Reviewer PfEp**
> >
> > Question 1 (effective complexity). The hidden dimension of OLMo-1b is 2048. However, we emphasize that the notion of ``effective complexity'' in the Park et al. [1] and Marconato et al. [2] papers corresponds to the rank of the single-token logit matrix, which is different from our setting where we consider the extended logit matrix (see Section 1.1 and Definition 2.2). Thus, we should not expect that using a rank-2048 approximation we would recover the distribution of the true model.
> >
> > As you observe, the key difference is that in our setting, the embeddings of histories can predict the distributions of entire futures, whereas in [1,2], the embeddings of histories only predict the distribution of the next token.
> > Thus, in the setting of [1,2], to actually generate more than one token from the model, we would have to recompute the embedding after adding on each token.
> > This new embedding could be a completely arbitrary function.
> >
> > Question 2 (principal angles). Please see the earlier explanation. If you have additional questions, we are happy to explain further.
> >
> > Question 3 (relational linearity). Thanks for pointing out the connection to relational linearity.  Indeed your observation is correct that the ISAN model essentially implies something more general, that any sequence of tokens can be viewed as a linear relation applied to the embedding of the history before it. We will add discussion of this point in the paper.

---

> > > ### Comment · Reviewer_PfEp · 2025-11-24
> > > **Reply**
> > >
> > > Thank you for going through the points I raised.
> > >
> > > Point 2: thank you for the clarification!
> > >
> > > Point 3: thank you for the explanation. Can you confirm that we expect different those decays because A and A' are used (so it depends on the choice of r in use)?
> > >
> > > Point 4: ok, maybe it is worth saying it in the text
> > >
> > > Point 5: interesting idea. can you test the efficacy, time permitting?
> > >
> > > Point 6:I understand your point about the ISAN model. My only concern was about the space it was devoted to it and that it feels a bit isolated part of the paper. I suggest better integrating that part.
> > >
> > > Q1: Would it make sense in your opinion to think about a notion of effective complexity for the histories futures setup? If so, what is it measuring?
> > >
> > > Q3: Thank you, I believe that is an interesting connection to other interpretable properties of LLMs

---

> > > > ### Author Response · Authors · 2025-11-27
> > > >
> > > > Thank you for your reply:
> > > >
> > > > Point 3: yes, the decay depends on the choice of the rank r of A, A'.
> > > >
> > > > Point 5: For generic sentences, the experiments already demonstrate in some sense that splitting the history does work, because we could think of the original history as $h_1$ and the first few generated tokens as $h_2$. Then note that the additional tokens that are generated remain coherent with the entire history $h_1 \circ h_2$. We believe a more comprehensive evaluation including e.g., jailbreaking/safety-focused benchmarks, is a great direction for follow-up work.
> > > >
> > > >
> > > >
> > > > Q1: Roughly speaking, the notion of "effective complexity" corresponding to our setup would refer to the rate of decay of the approximation error for a rank-r matrix, for various values of r (e.g., what is plotted in Figure 1a). This complexity would be measuring the dimension of embedding needed to (approximately) "linearly generate" an entire continuation, rather than just a single token.

---

> > > > > ### Comment · Reviewer_PfEp · 2025-11-27
> > > > >
> > > > > Thank you for additional clarifications and comments. I congrat the authors for the nice rebuttal and very nice paper.
> > > > >
> > > > > I'll keep my positive score.

---

### Official Review · Reviewer_7Jjj · 2025-11-01

**Soundness:** 3
**Presentation:** 2
**Contribution:** 3
**Rating:** 6
**Confidence:** 2

**Summary:**

This paper proposes to analyze the low-dimensionality of LLMs' generation process by studying a curious construction called the extended logit matrix, which consists of the log-probability of the topK predicted tokens $\log P(Z|H\circ F)$, where $H$ and $F$ are segments of text randomly sampled from a certain source. They found low-dimensional behaviour in the resulting logit matrix, which suggests linear dependence - the predicted logits to some prompts could be closely approximated as an linear combination of predicted logits of other, statistically unrelated prompt. To prove this point, they introduce LINGEN, a linear generation algorithm that reconstructs the output distribution for a target prompt by linearly combining logits from unrelated prompts via a weighting vector $v$ obtained via linear regression. Surprisingly, this produces coherent text—even when the basis prompts are nonsense—revealing strong linear constraints in LLM's autoregressive generation.

**Strengths:**

- The framework is architecture agnostic: Studies LLMs through their output distribution geometry rather than internal activations.

- The analysis of extended logit matrix is relatively straightforward and tractable via matrix decomposition algorithms.

- The fact that one could infer LLM generation of a target prompt using unrelated or even non-sensical prompts is very surprising.

**Weaknesses:**

- While the low-rank property is convincingly demonstrated, the paper stops short of providing a theoretical account for why transformer models should exhibit such linearity in distribution space. However, I do not hold this against the paper since the paper is already phenomenologically rich and self-contained.
- The explanation of how the extended logit matrix is constructed, as well as its motivation, is quite abstruse. I think the paper could benefit from a cartoon illustration of the structure of the extended logit matrix, as well as the LINGEN algorithm.
- Although LINGEN reproduces plausible text, the qualitative examples remain small-scale, and quantitative evaluations of long-horizon coherence or perplexity are limited.

**Questions:**

- Why is $v$ fit in log-probability space rather than pre-softmax logits, and how sensitive are results to this choice?
- Could the observed low-rank structure be derived from, or correspond to, linear relationships in the model’s internal hidden states (e.g., out embedding)?

---

> ### Author Response · Authors · 2025-11-19
> **Response to Reviewer 7Jjj**
>
> Thank you for the feedback on the extended logit matrix. We will add a cartoon illustration of the extended logit matrix and LINGEN algorithm and flesh out how it arises very naturally from thinking about linear embeddings of histories and futures as in Section 1.1.
>
> In terms of the scale of the experiments, the main goal of this paper is to develop a method and show that it works in principle. We agree that a more extensive, larger-scale evaluation would be a natural and promising next step but may require significantly more computational resources. Nevertheless, we believe that the results, in particular the consistency of the power law scaling, give evidence that suggest the techniques should scale.
>
> In fact, for LINGEN specifically, generating more tokens becomes ``easier" because the full sequence of generated tokens gets appended to all of the basis histories. This is also reflected in the plots, e.g. Fig 1(b), where the KL error per token is actually smaller at later tokens in the sequence.  We do think that it would be interesting to try to run LINGEN for a longer target history and to test whether it can do long-context recall.
>
> Question 1: log-probability vs pre-softmax logits. The normalization doesn't seem to matter. We choose to mean-center the logits because this plays nicely with the theory, but the results all appear essentially the same with log probabilities or raw (pre-softmax) logits.
>
> Question 2: linear relationships in the model's internal states. We did run a similar experiment where we replace the vector of logits in the extended logit matrix with the last hidden state (also subsampling its entries) and we observe a very similar power law decay in the singular values.  Understanding how this structure relates to the internal layers seems quite complex but would be an interesting question for future work.

---

> > ### Comment · Reviewer_7Jjj · 2025-11-27
> >
> > I thank the authors for the comprehensive reply and look forward to the updated manuscript. I have updated my score.

---

### Author Response · Authors · 2025-12-03
**Summary for New Area Chair**

We thank the reviewers for their insightful comments and productive discussion. To recap the changes that happened (in addition to the discussion that is visible):

1. We addressed reviewer comments on presentation-- we added

    a. Illustrations and motivation for the extended logit matrix and LINGEN algorithm

    b. Additional explanation of principal angles and how to interpret the implications for "transfer" (how linear relationships between histories remain invariant even if you change the set of futures or the model)

    c. Additional discussion on limitations of LINGEN -- namely needing to query the target history many times, and potential ways to get around this e.g. by splitting the target history

    d. Comment on the connection between ISAN and a general form of relational linearity

2. Reviewer 7Jjj updated their score to 8

---

### Meta-Review · Area_Chair_Qb4e · 2026-01-05

**Summary:**

This paper provides extensive empirical evidence and analysis showing that modern LLMs' logit matices exhibit a low-rank structure, and such a structure is transferable across different (even nonsensical) prompts and models. This provides a model-agnostic model for analyzing the innerworkings of modern LLMs that strikes an appealing balance between fidelity to practice and feasibility to rigorous analysis.

Reviewers unanimously appreciate the novelty of the proposed framework and that it leads to interesting or surprising consequences (e.g., the proposed LINGEN algorithm). While reviewers have raised some concerns regard presentation/experiments/detailed demonstrations of further implications (see Reviewer Concerns), the authors have addressed most of them during the rebuttal, and I very much agree with the reviewers that the current finding and analysis is already interesting, fundamental, and potentially paves an avenue for many future directions.

I therefore recommend clear acceptance.

**Reviewer Concerns:**

Reviewers raised concerns on the presentation clarity of the paper, e.g., how the extended logit matrix is constructed, and lack of details on the proposed method LINGEN and experiments. Another main concern is that the paper did not show empirical results on larger-scale sentence generation/LLM safety experiments. The authors have addressed the former concern during the rebuttal and acknowledge the latter, leaving it as future work, which I feel acceptable given that the current results are already comprehensive and significant.

**Reviewer Scores:**

Reviewer 7Jjj explicitly mentioned that they intended to update the score from 6 to 8; reviewers PfEp and L59i explicitly mentioned the intention to keep their original scores of 8.

---

### Decision · Program_Chairs · 2026-01-26

Accept (Oral)